# Spatiotemporal interplay between multisensory excitation and recruited inhibition in the lamprey optic tectum

**Andreas A Kardamakis‡, Juan Pérez-Fernández‡, Sten Grillner***

Department of Neuroscience, Karolinska Institute, Stockholm, Sweden

**Abstract** Animals integrate the different senses to facilitate event-detection for navigation in their environment. In vertebrates, the optic tectum (superior colliculus) commands gaze shifts by synaptic integration of different sensory modalities. Recent works suggest that tectum can elaborate gaze reorientation commands on its own, rather than merely acting as a relay from upstream/forebrain circuits to downstream premotor centers. We show that tectal circuits can perform multisensory computations independently and, hence, configure final motor commands. Single tectal neurons receive converging visual and electrosensory inputs, as investigated in the lamprey - a phylogenetically conserved vertebrate. When these two sensory inputs overlap in space and time, response enhancement of output neurons occurs locally in the tectum, whereas surrounding areas and temporally misaligned inputs are inhibited. Retinal and electrosensory afferents elicit local monosynaptic excitation, quickly followed by inhibition via recruitment of GABAergic interneurons. Multisensory inputs can thus regulate event-detection within tectum through local inhibition without forebrain control.

**\*For correspondence:** sten. grillner@ki.se

‡These authors contributed equally to this work

**Competing interests:** The authors declare that no competing interests exist.

## Introduction

Sensorimotor circuits have been studied in a wide range of biological organisms in pursuit of identifying the operational principles that govern the integration of sensory information from different modalities for the generation of goal-directed behavior. The optic tectum (superior colliculus in mammals), has received particular attention for its distinct role in orienting behavior, i.e. the control of orienting and avoidance gaze movements (*Dean et al., 1989*; *Moschovakis et al., 1996*; *Basso and Wurtz, 1997*; *Sparks, 2002*), through the integration of different sensory modalities (which are species-dependent) like vision, auditory and electroreception (*Bodznick and Northcutt, 1981*; *Meredith and Stein, 1986*; *Wallace et al., 1996*, *Gingras et al., 2009*). Although combining multiple sensory inputs has been proposed to increase the reliability of event detection in the environment (*Ernst and Banks, 2002*; *Fetsch et al., 2009*), little is known about the neural mechanisms underlying this integration.

Studies of the superior colliculus have established a set of empirical principles that place constraints on the spatial and temporal dimensions underlying multisensory integration (*Stein and Stanford, 2008*). In particular, extracellular activity correlated to gaze shift execution would increase with spatiotemporally congruent cues from two senses or decrease with spatially disparate and/or temporally asynchronous cues (*Meredith et al., 1987*; *Meredith and Stein, 1996*; *Kadunce et al., 1997*; *Recanzone, 2003*). Conceptual and computational models of multisensory integration have proposed the existence of an inhibitory mechanism to account for these effects (*Rowland et al., 2007*; *Alvarado et al., 2008*; *Ursino et al., 2009*; *Ohshiro et al., 2011*; *Miller et al., 2015*), although they have so far remained hypothetical. The goal of this study is to determine the cellular and synaptic mechanisms embedded within the optic tectum that control multisensory integration.

**eLife digest** Many events occur around us simultaneously, which we detect through our senses. A critical task is to decide which of these events is the most important to look at in a given moment of time. This problem is solved by an ancient area of the brain called the optic tectum (known as the superior colliculus in mammals).

The different senses are represented as superimposed maps in the optic tectum. Events that occur in different locations activate different areas of the map. Neurons in the optic tectum combine the responses from different senses to direct the animal's attention and increase how reliably important events are detected.

If an event is simultaneously registered by two senses, then certain neurons in the optic tectum will enhance their activity. By contrast, if two senses provide conflicting information about how different events progress, then these same neurons will be silenced. While this phenomenon of 'multisensory integration' is well described, little is known about how the optic tectum performs this integration.

Kardamakis, Pérez-Fernández and Grillner have now studied multisensory integration in fish called lampreys, which belong to the oldest group of backboned animals. These fish can navigate using electroreception – the ability to detect electrical signals from the environment. Experiments that examined the connections between neurons in the optic tectum and monitored their activity revealed a neural circuit that consists of two types of neurons: inhibitory interneurons, and projecting neurons that connect the optic tectum to different motor centers in the brainstem.

The circuit contains neurons that can receive inputs from both vision and electroreception when these senses are both activated from the same point in space. Incoming signals from the two senses activate the areas on the sensory maps that correspond to the location where the event occurred. This triggers the activity of the interneurons, which immediately send 'stop' signals. Thus, while an area of the sensory map and its output neurons are activated, the surrounding areas of the tectum are inhibited.

Overall, the findings presented by Kardamakis, Pérez-Fernández and Grillner suggest that the optic tectum can direct attention to a particular event without requiring input from other brain areas. This ability has most likely been preserved throughout evolution. Future studies will aim to determine how the commands generated by the optic tectum circuit are translated into movements.

In many vertebrates, vision and electroreception are the spatial senses that are used to localize predators and prey in their immediate environment, whereas other species also rely on auditory, somatosensory, infrared, echolocation and/or magnetic systems. These modalities are represented within the optic tectum and are used for orienting and avoidance behaviors (*Hartline et al., 1978*; *Semm and Demaine, 1986*; *Valentine and Moss, 1997*) in vertebrates extending from lampreys to primates (*Wurtz and Albano, 1980*; *Nieuwenhuys and Nicholson, 1998*; *Saitoh et al., 2007*; *Jones et al., 2009*; *Asteriti et al., 2015*; *Kardamakis et al., 2015*). We have previously shown in the lamprey that site-specific stimulation across the deep layer of the optic tectum gives rise to eye-head gaze shifts of given amplitude and direction, thus, showing the existence of a motor map (*Saitoh et al., 2007*). We now show that visual and electroreceptive inputs are integrated in the same deep layer neurons of the optic tectum, which provide the output to different brainstem centers. Projecting output neurons, as well as local interneurons, receive monosynaptic excitatory input from both sensory afferent pathways and also disynaptic inhibition triggered by the same afferents. This applies if the two signals are activated from the same point in space, whereas signals from surrounding areas provide only inhibition (*Kardamakis et al., 2015*). The membrane properties of the tectal output neurons ensure the temporal integration of bimodally triggered excitatory and inhibitory currents. Due to the highly conserved organization of the optic tectum (*Nieuwenhuys and Nicholson, 1998*; *Saitoh et al., 2007*; *Jones et al., 2009*; *Asteriti et al., 2015*; *Kardamakis et al., 2015*), we anticipate that the mechanisms of integration of two senses on single output neurons, as demonstrated here, may also apply to other vertebrates.

# Results

## Spatial organization of sensory inputs and motor output in the optic tectum

The full projection patterns of tectal efferents to the brainstem and incoming visual and electroreceptive afferents arising from the retina and the octavolateral area in the intact brain was examined (*Nieuwenhuys and Nicholson, 1998*; *Jones et al., 2009*; *Ronan and Northcutt, 1987*) (*Video 1*). For this, we used a tracer injection into the optic tectum followed by passive CLARITY-optimized light-sheet microscopy (*Tomer et al., 2014*). Tectal efferents arise from neurons in the deep layer (DL) and project to the brainstem, where they make direct synaptic contacts onto the somata of reticulospinal (RS) neurons in the middle rhombencephalic reticulospinal nucleus (MRRN). These projecting neurons in the deep layer of the optic tectum (or superior colliculus) control gaze movements (*Sparks, 2002*; *Robertson et al., 2006*, *Saitoh et al., 2007*; *Jones et al., 2009*; *Kardamakis et al., 2015*) and will herein be referred to as 'output neurons'.

The optic tract, which carries retinal information, and the octavolateral tract, which carries electrosensory information to the optic tectum can be seen distinctly in *Video 1*. By injecting two different tracers into the contralateral optic nerve and the contralateral octavolateral area (*Figure 1A-i*), we revealed that the two sensory modalities have spatially segregated termination zones with minimal overlap (retina in red, octavolateral in green; n = 3, *Figure 1A-ii*). Retinal input targets the superficial layer, while electroreceptive afferents project to the intermediate layer. The deep layer (DL, visible with the Nissl stain; in blue) contains neurons that have dendritic arbors extending through the intermediate and into the superficial layers (*Figure 1A-iii*), as revealed by intracellular staining of output neurons that were prelabeled following retrograde tracer injections into the MRRN. Effectively, they exhibit the optimal morphological structure to support sampling of layer-specific inputs carrying visual and electrosensory information.

## Visual and electrosensory integration with local inhibition

To test the mechanisms underlying the integration of the two sensory modalities in tectal output neurons, we first used an intact preparation that enabled extracellular monitoring of neural activity from the deep layer, while visual and electrosensory inputs could be stimulated separately or in combination with coordinated spatial and temporal alignment (*Figure 1B*; *see* Materials and methods). Separate stimulation of visual inputs with brief pulses of light (500 ms duration) or electroreceptive inputs activated by electrical pulses (30 ms duration) led to bursts of activity in the output neurons (*Figure 1C-i,ii*, control traces). When the two sensory inputs were delivered in an overlapping temporal sequence, there was a marked enhancement of deep layer neural activity (*Figure 1C-iii*, control; n = 6). To achieve an enhancement, the two stimuli had to be delivered to the same parts of the visual and electroreceptive fields, respectively.

To quantify the impact of stimulus strength on bimodal integration, we systematically varied the extent of visual and electrosensory activation by means of local electrical microstimulation of the retina and the anterior lateral line nerve, respectively (for further information, *see* Materials and methods). To mimic the spatial resolution of visual stimuli, we activated the retinal area that coincided with the receptive field center of our recording sites. Stimulation of the rostral branches of the anterior lateral line nerve would simulate electrosensory stimuli incoming from the frontal regions (*Ronan and Northcutt, 1987*). Varying

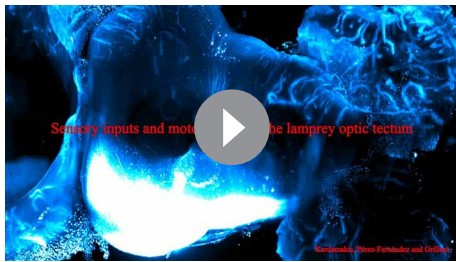

**Video 1.** Sensory inputs and motor output in the lamprey optic tectum in the intact brain using COLM. The visual input through the optic tract (OpT) to the optic tectum (OT), and the electrosensory afferents from the octavolateral area (OLA) are shown after a neurobiotin injection in the OT, in a cleared brain using the method CLARITY. The motor output can be also followed from the deep layer to the reticulospinal cells in the brainstem. The brain is shown from a dorsal view and in the bottom-left corner a schematic in a sagittal view can be seen. A moving bar indicates the approximate region shown at each moment. Areas of interest are annotated through the duration of the movie.

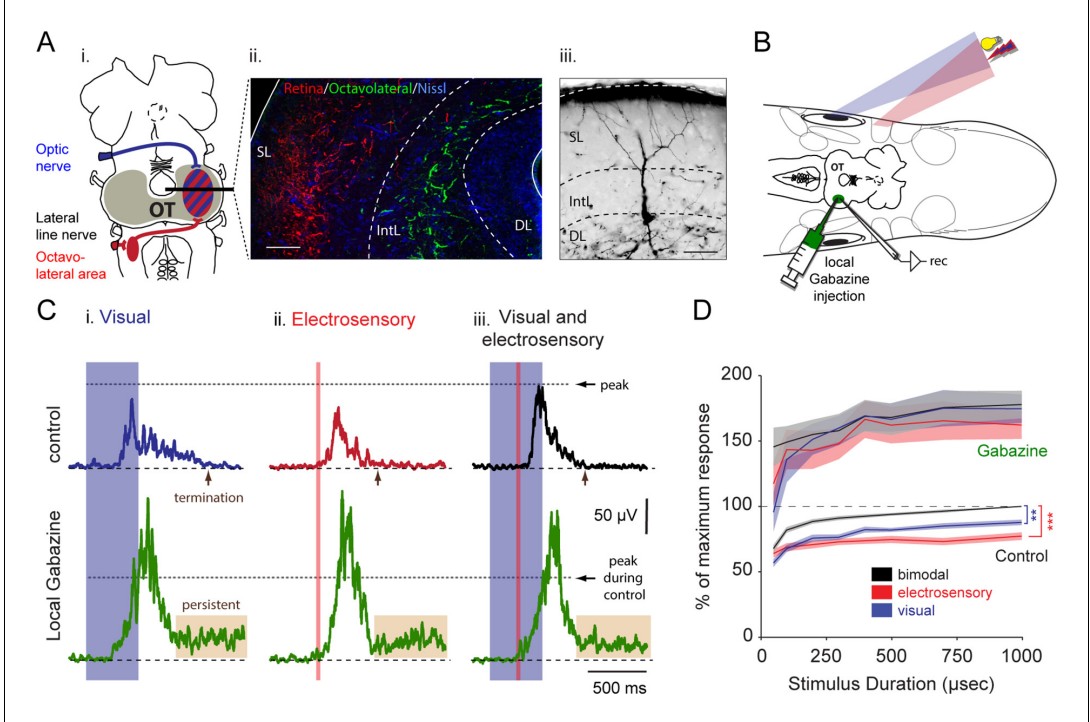

**Figure 1.** Integration of vision and electroreception in the deep layer of the lamprey optic tectum. (A) *Inset* i: Schematic of the lamprey brain showing the visual (blue) and electrosensory (red) afferents targeting the optic tectum (OT). *Inset* ii: Photomicrograph of the optic tectum in a transversal view showing the retinal afferents reaching the most superficial layers (red), and the octavolateral fibers innervating the intermediate layers (green). *Inset* iii: Morphology of an output neuron in the deep layer retrogradely labeled following a tracer injection in the middle rhombencephalic reticulospinal nucleus (MRRN) and filled intracellularly with Neurobiotin while performing whole-cell recordings. Output cells extend their dendrites to the intermediate and superficial layers where the electrosensory and the visual inputs enter and terminate, respectively. *Abbreviations*: SL, superficial layer; IntL, intermediate layer; DL, deep layer. Scale bars: *Inset* ii, 100 μm; *Inset* iii, 50 μm. (B) Experimental settings for performing extracellular recordings during multisensory integration in the optic tectum. Dorsal view of the preparation, including the brain, the eyes and electrosensory areas (depicted by the skin patches; for more information see *Bodznick and Preston [1983]*), while driving output activity with light and electrical stimuli that are spatiotemporally aligned in the immediate surrounding. *Abbreviations*: rec: extracellular recording electrode. (C) Rectified local field potentials obtained from visual (inset i), electrosensory (inset ii) and bimodal sensory activation (inset iii). Upper traces show sensory stimulation before (black), and after local application of 10 μM gabazine (green). Horizontal dotted lines illustrate the level of peak activity during control. (D) Sensory response against stimulus duration (50–1000 μs) for visual, electroreceptive and bimodal activation, with and without local inhibition. The integral under the curve of rectified local field potentials, as those shown in C, is plotted on the y-axis and normalized to the maximum bimodal response measured during control (*n = 13*). Paired *t*-test gave statistical significance as indicated (**p<0.01, ***p<0.001).

The following figure supplement is available for figure 1:

**Figure supplement 1.** Actual responses against the predicted arithmetic sum of unisensory responses.

stimulus durations (*Figure 1D*) ranging from 50 μs to 1 ms were used to drive increasing tectal responses in the deep layer, which were measured from their rectified activity. The lower curves in *Figure 1D* illustrate the normalized effect that stimulus duration had on visual, electrosensory and combined responses during physiological conditions (*n = 13*). Unisensory inputs generated up to ~55% of the maximal bimodal response across the entire stimulus range. During bimodal activation, the combined response significantly exceeded the unimodal responses (visual-bimodal: p<0.01; electrosensory-bimodal: p<0.001 between 100–1000 μs; *Figure 1D*, black control trace).

Recently, we have shown how on-receptive field visual responses in tectal output layer neurons can be suppressed, via the local inhibitory system, by the presence of multiple visual stimuli located at disparate positions in the visual field (*Kardamakis et al., 2015*). We now show that this unisensory response suppression can also be achieved by using multiple stimuli of different sensory modalities (*Figure 2A*). Once a tectal region responsive to a local electroreceptive stimulus was established

(red trace, *Figure 2B*), we were able to suppress the magnitude of the on-response (red trace) by delivering a stimulus to an off-response region of the retina (black trace). To confirm that the lack of activity in response to the visual stimulation (blue trace) was not due to unsuccessful stimulation, we performed control recordings in other tectal regions, as well as in downstream brainstem regions. An average response reduction of ~75% was observed during spatially misaligned cross-modal sensory stimulation (*Figure 2C*; red for electrosensory and black for bimodal stimulation; n = 5). By contrast, we were able to suppress by a negligible amount of only ~3% (*data not shown*) when the opposite combination of sensory modalites were used, i.e. on-responses to visual and off-response to electrosensory stimuli. The underlying cause that gives rise to this asymmetry remains unclear.

## Multisensory integration after blockade of local inhibition

To evaluate the role of local inhibition, we microinjected the GABA$_A$-receptor antagonist gabazine (10 μM) in the area of the recording site (*Figure 1B*). We observed a marked increase in the magnitude of the response during drug application (*Figure 1C*, bottom traces). The responses to visual, electrosensory or the combination of both were now virtually identical. Without inhibition, neurons in the deep layer became irresponsive to the following incoming stimuli for a refractory period that exceeded 30 s (n = 6).

When gabazine was microinjected (*Figure 1C,D*), the responses increased by approximately 80–100% compared to their original magnitude throughout the stimulus durations (for visual: 85.5 ± 13.7%; electrosensory: 97.4 ± 14.4%; for bimodal: 82.4 ± 13.0, Means ± SD; n = 13) and were accompanied with a significantly higher degree of variance. At the same time, the relationship between unimodal and bimodal responses quickly deteriorated (no significant differences) with visual inputs dominating bimodal activation. To visualize the extent of the enhancement of unisensory response during bimodal activation, we plotted the actual data in control conditions and after the application of Gabazine as a function of the predicted arithmetic sums of each unisensory response (*Figure 1— figure supplement 1*). Our data thus suggest that tectal inhibition is critical to ensure response enhancement and a stable relationship between response amplitudes to unimodal inputs.

## Bimodal inputs enhance the probability of neural activation

Given that the role of inhibition is critical for integrating the two unimodal inputs (*Figure 1C,D*), a central issue was to determine the inhibitory mechanisms that shape the synaptic integration at the

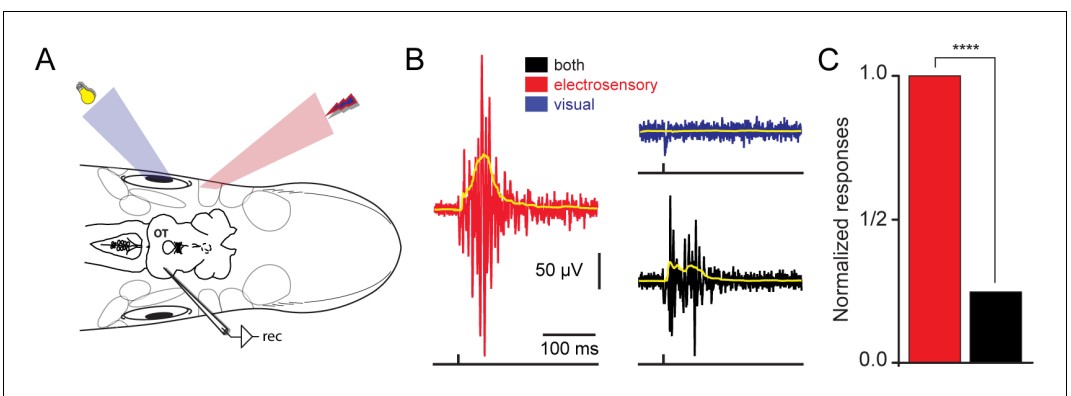

**Figure 2.** Spatially misaligned stimuli give rise to response reduction. (**A**) Using the experimental strategy described in *Figure 1B*, we applied spatially disparate visual and electrosensory stimuli while recording responses in the contralateral optic tectum. (**B**) Local field potentials in response to electrosensory stimulation (red trace), were drastically reduced when a different region of the tectal map was visually stimulated (blue trace). The responses when combining both sensory modalities are shown in black. The yellow traces on top show the rectified signals. (**C**) Plot showing the normalized responses for electrosensory activation (red), before and after simultaneously stimulating a visual off region (black). For each animal (n = 5), stimulation was repeated throughout 10 sweeps. Responses were quantified as the area under the rectified signal and normalized to the maximum. Paired *t*-test gave a statistical significance of ****p<0.0001.

level of the single output neuron. To address this, we performed whole-cell and cell-attached recordings from these cells by exposing the deep layer of the optic tectum in a slightly oblique sagittal midbrain section (~500–600 µm) that maintains the mesencephalon and parts of the diencephalon and rhombencephalon directly adjacent to the midbrain (*Figure 3A*). We could then selectively stimulate both the optic tract and the fiber bundle that exits the octavolateral nucleus (*Figure 3A*), while recording from tectal output neurons.

To evaluate the reliability of bimodal integration, we initially recorded responses in cell-attached mode to ensure that the cell membrane and cytoplasm would remain intact, while delivering sustained presynaptic stimulation of the sensory afferent pathways (20 pulses/10 Hz; *Figure 3B*, left). We then quantified the likelihood of their all-or-none responses to each impulse (n = 6; *Figure 3B*, right). Bimodal activation resulted in a higher spiking probability of single stimulus-locked action

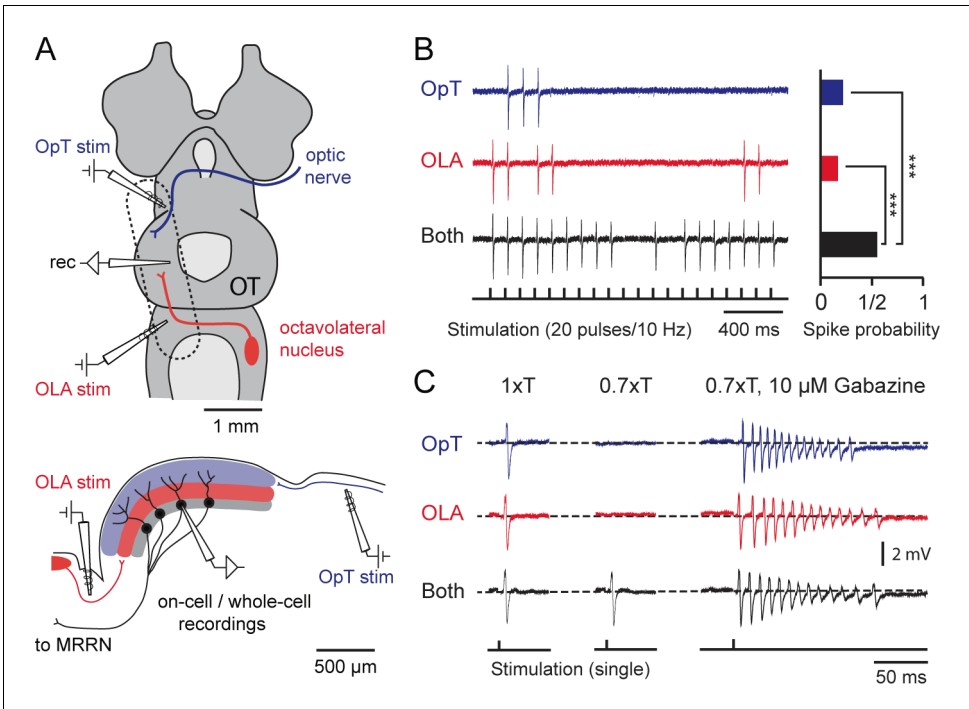

**Figure 3.** Output neurons receive visual and electrosensory inputs. (**A**) (Top) Schematic of the lamprey brain in a dorsal view showing the sensory afferent input to the optic tectum. Dotted rectangle shows the brain region of interest (sectioned and shown in sagittal view; Bottom) illustrating the settings for performing cell-attached and whole-cell recordings in tectal output neurons while stimulating the visual (via the optic tract, OpT) and electrosensory (via the octavolateral area, OLA) inputs. *Abbreviations*: OpT, optic tract; OLA, octavolateral area; OT, optic tectum; stim: stimulation electrode; MRRN, middle rhombencephalic reticulospinal nucleus. (**B**) (Left) Tectal output cell recordings driven by visual (OpT), electrosensory (OLA) and bimodal (both) inputs in cell-attached configuration (stimulation train of 20 pulses at 10 Hz; low threshold 1–10 µA). Time-locked action potentials can be observed in response to an impulse from either sensory pathway with enhanced excitability occurring during bimodal activation. (Right) Quantification of spike responses to unimodal and bimodal activation. A probability of unity indicates that an output neuron responds to all given impulses throughout 10 sweeps (n = 6). Paired *t*-test gave a statistical significance of ***p<0.001. (**C**) (Left) Action potentials discharged in response to unimodal and bimodal input at stimulation intensity threshold (T). (Middle) Bimodal inputs drive suprathreshold responses in output neurons, with unimodal inputs failing to elicit spiking by adjusting T to a 70% of its initial value (n = 6). (Right) Unimodal and bimodal inputs yield equalizing effects with a rapid discharge of action potentials (at 0.7T) when blocking inhibition with bath application of 10 µM of Gabazine (n = 3).

The following figure supplement is available for figure 3:

**Figure supplement 1.** Unimodal and bimodal sensory activation with and without inhibition – whole-cell recordings.

potentials (statistical differences of p<0.001). Unimodal stimuli were usually able to drive output neurons to discharge one time-locked action potential with stimulation intensity thresholds (1xT) ranging from 1–10 μA (*Figure 3C*, left panel). Reducing T by 30% (0.7xT) would elicit a suprathreshold response only when both pathways were engaged (*Figure 3C*, middle panel). This combined action of weaker stimuli leading to stronger bimodal responses is a key feature underlying multisensory integration. This enhancement is completely lost in the absence of local GABAergic inhibition, where output neurons rapidly fired multiple action potentials even in response to normally ineffective stimuli as in the case of 0.7xT (*Figure 3C*, right panel; see also *Figure 1C* and *Figure 3—figure supplement 1*).

## Tectal output cells are temporal integrators

To determine the integrative properties of tectal output cells, we first established their active membrane properties with whole-cell voltage measurements in response to applied current steps (*Figure 4A-i,ii*). Notably, these cells have long membrane time constants (98.8 ± 11.5 ms, *n = 32; see* Materials and methods) due to their high input resistance (0.95 ± 0.24 GOhm, *n = 32*). This property facilitates the temporal summation of excitatory currents and ensures that distal visual inputs (in superficial layer) and more proximal electrosensory inputs (in intermediate layer) will yield functionally similar responses (i.e. amplitude and temporal characteristics) at the final integration site in the soma (in the deep layer; see *Figure 1C*, top traces). Furthermore, output neurons also discharge action potentials with a regular firing pattern in response to stepwise increases of applied current, and display a continuous and linear frequency-current relationship in the lower range of current injections (*Figure 4A-iii*; slope: 0.53 ± 0.05 spikes/pA, *n = 32*). This was further corroborated in experiments in which the action potentials were blocked intracellularly by QX-314 (blocking fast sodium channels) and the suprathreshold membrane fluctuations in response to positive current steps were monitored (*Figure 4A-ii*). All neurons that were recorded had a linear input-output

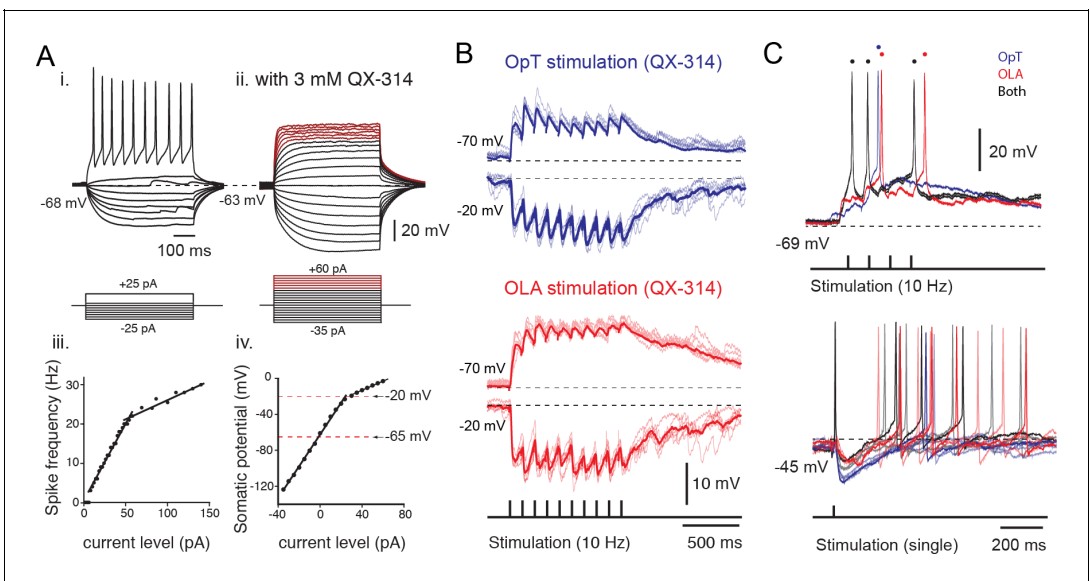

**Figure 4.** Sensory excitation and tectal inhibition are integrated by output neurons. (A) *Inset* i. Voltage responses to depolarizing and hyperpolarizing 500 ms current steps of 2 pA per step, elicited from rest at −68 mV. *Inset* iii. Spike frequency is plotted against current level. *Inset* ii. Voltage responses to depolarizing and hyperpolarizing 500 ms current steps of 5 pA per step, elicited from rest at −63 mV in the presence of QX-314 intracellularly (3 mM). Voltage traces marked in red belong to the shallow slope in the I-V plot indicative of a drop in DC impedance. *Inset* iv: Plot of somatic potential against current level. (B) Whole-cell recordings of excitatory and inhibitory postsynaptic potentials evoked in output cells that were held at −70 mV and −20 mV, respectively, in response to repetitive stimulation of OpT (top traces; blue) and OLA (bottom traces; red) afferents at 10 Hz, in the presence of fast-sodium channel blocker QX-314 in the recording pipette. Average traces are shown as thicker lines. (C) (Top) Output neuron responses to sensory stimulation (train of 4 pulses at 10 Hz) of OpT (blue traces), OLA (red traces) and bimodal (black traces) from rest at −69 mV. (Bottom) Output responses recorded when holding near threshold (at −45 mV with positive current injection) reveals evoked inhibitory postsynaptic potentials to single impulses that also persist during bimodal integration.

relationship across a wide range of membrane potentials (−120 to −20 mV; *Figure 4A-iv*) and would, thus, smoothly encode the strength of similar sensory inputs into output firing frequency.

## Sensory excitation triggers the local inhibitory system

To determine the synaptic underpinnings of multisensory integration, we performed whole-cell recordings while applying a train of stimuli (at 10 Hz) to the retinal (OpT) and octavolateral (OLA) afferent pathways (*Figure 4B*). We found that excitatory postsynaptic potentials were evoked onto the same output neurons. A concurrent inhibitory postsynaptic potential was also revealed when the membrane potential was held at −20 mV (upper range of the linear domain of their V-I curve) in current clamp mode. This inhibition is feedforward and is produced mainly disynaptically by projections of tectal interneurons onto output neurons (*Kardamakis et al., 2015*; *see* below).

In a subset of neurons (4/6) studied in cell-attached configuration (as shown in *Figure 3*), the membrane was later ruptured allowing us to enter into whole-cell configuration and measure the combined excitatory and inhibitory action of stimulating both visual and electrosensory pathways (*Figure 4C*, black traces). Consistent with extracellular (*Figure 1C*) and cell-attached recordings (*Figure 3A*), the output neurons reach threshold more reliably and faster when both sensory modalities are present (black trace in *Figure 4C*; resting at −69 mV; top traces), even though unisensory stimuli can eventually drive spiking (*Figure 4C*; retinal in blue; electrosensory in red). Notably, the inhibition persisted not only during unimodal but also during bimodal stimulation. This can be seen by the long-lasting inhibitory postsynaptic potential immediately following the first action potential, which is visible when holding the neurons near threshold (*Figure 4C*, bottom traces). Together, these data indicate that the membrane properties of output neurons are able to support the smooth integration of sensory-evoked excitation coupled to triggered inhibition.

## Sensory excitation and local inhibition are fully integrated by output neurons

To improve the isolation of synaptic conductances underlying this sequence between excitation and inhibition during unimodal and bimodal integration, we blocked action potentials with the fast sodium-channel blocker QX-314. In voltage-clamp, the synaptic currents evoked by unimodal stimulation summate during bimodal stimulation (*Figure 5A*). Whole-cell current recordings (retinal are blue traces; octavolateral are red traces) began with a direct excitatory current that was followed ~5–10 ms later by an intense inhibitory current (*see* also *Kardamakis et al., 2015*), when the somatic potential was held at the reversal potentials for inhibition (−65 mV) and excitation (0 mV), respectively. When co-stimulating both afferent pathways, excitatory postsynaptic current and inhibitory postsynaptic current amplitudes increased in direct correspondence to the unimodal-evoked responses (*Figure 5A,B*). On a trial-to-trial basis, the statistical difference in the magnitude of evoked EPSC and IPSC currents by bimodal and visual or electroreceptive afferent stimulation was significant (visual-bimodal: p=0.008; electrosensory-bimodal: p=0.006; one-way ANOVA), but not between those evoked during visual and electrosensory (p=0.99). The same was true for the evoked IPSCs (visual-bimodal: p=0.006; electrosensory-bimodal: p=0.0009; both visual and electrosensory against bimodal: p=0.24). Notably, no statistical significance (p=0.85 for EPSCs and p=0.92 for IPSCs) was detected when comparing the mean differences between the sum of unisensory-evoked and bimodal-evoked currents (*Figure 5B*; EPSCs, visual: 28.63 ± 5.6 pA, electrosensory: 28.4 ± 5.85 pA, both: 56.27 ± 9.11, n = 12; IPSCs, visual: 24.04 ± 6.15 pA, electrosensory: 33.7 ± 14.82 pA, both: 55.22 ± 15.79 pA, n = 7). As predicted by their membrane properties, output neurons are able to temporally integrate the synaptic currents evoked by individual unisensory inputs summate to generate an enhanced bimodal product.

This temporal integration of excitation and inhibition has a differential impact on the membrane potential. *Figure 5C* shows the quantification of excitatory postsynaptic potentials (holding at −65 mV; a representative example of evoked EPSPs is shown in *Figure 6B*) and inhibitory postsynaptic potentials (holding at −20 mV) elicited by separate and combined (black) stimulation of the octavolateral (red) and optic tract (blue). After a brief transient phase (usually after three pulses), both EPSPs and IPSPs peak constant values during further repetitive stimulation (*Figure 5C*). Compared to unisensory stimulation, bimodal stimulation resulted not only in larger EPSPs (visual-bimodal: p=0.0001; electrosensory-bimodal; p=0.006; visual: 12.8 ± 1.4 mV, electrosensory: 10.5 ± 0.9 mV,

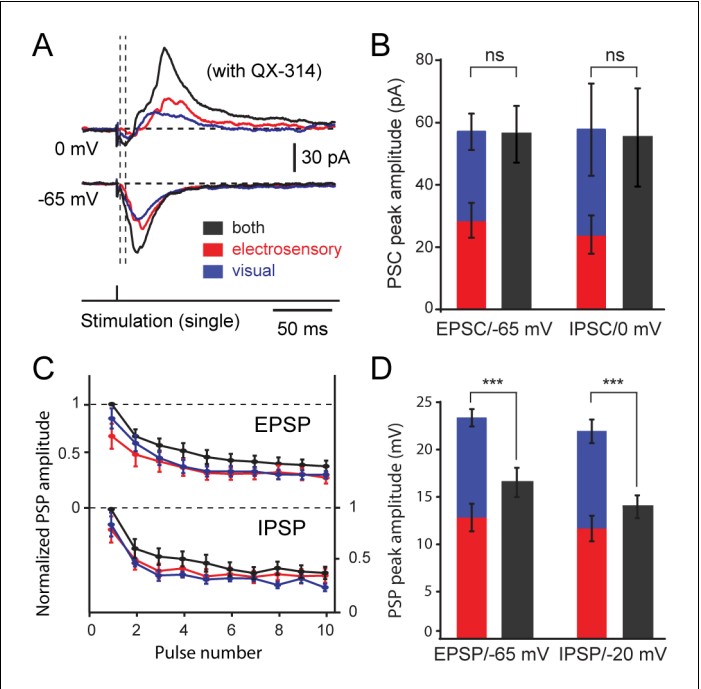

**Figure 5.** Excitatory and inhibitory postsynaptic currents and potentials evoked from visual and electrosensory inputs onto tectal output cells. (**A**) Inhibitory and excitatory postsynaptic currents (EPSCs) elicited by visual, electrosensory and bimodal stimulation in an output cell recorded in voltage-clamp at 0 mV to show inhibitory currents, and at −65 mV (equilibrium for chloride-mediated GABAergic inhibition) to show excitatory currents. Drop lines show the onsets of excitatory and inhibitory currents. (**B**) Quantification of peak amplitudes of postsynaptic currents elicited by OpT (blue) and OLA (red) stimulation summate linearly when compared to the bimodal stimulation (grey) in output cells (*n = 12*). *Abbreviation*: PSC, postsynaptic currents. (**C**) Quantification of postsynaptic potential (PSP) amplitudes in output cells evoked by sustained stimulation (10 pulses at 10 Hz) of the OpT (blue), OLA (red) and bimodal input (black) and recorded in current clamp mode. Values are normalized to the first PSP. Both excitatory postsynaptic potentials (EPSPs, upper curves) and inhibitory postsynaptic potentials (IPSPs, lower curves) decay during the first two to three impulses and reach steady-state thereafter. (**D**) Comparison of the first evoked EPSP and IPSP amplitudes (obtained from traces as shown above). In current clamp, however, EPSP and IPSP amplitudes do not summate in unimodal and bimodal conditions. QX-314 was applied in the pipette during these recordings. Data represented as Means ± SEM. One-way ANOVA was used to determine the p-value.

bimodal: 16.4 ± 1.5 mV, *n = 24*), but also in larger IPSPs (visual-bimodal: p=0.05; electrosensory-bimodal; p=0.02; visual: 11.7 ± 1.4 mV, electrosensory: 10.2 ± 1.2 mV, bimodal: 13.9 ± 1.2 mV, *n = 24*; *Figure 5D*)., The quick opposing action of feedforward inhibition curtails the temporal window that allows output neurons to integrate excitatory inputs, thus, restraining the impact that excitation has on membrane potential deflection. When bimodal inputs are used, excitatory currents summate and increase the resultant EPSP amplitude (*Figure 5C,D*) but are quickly quenched by an also stronger amount of inhibition.

To capture the dynamics underlying bimodal stimulus integration, we measured the synaptic responses in current clamp at varying holding potentials (usually at −65, −45, and −20 mV) by using QX-314 in the recording pipette and then calculated the synaptic conductances during the evoked unisensory and bimodal responses using conventional linear methods (*Monier et al., 2008*). The plot in *Figure 6A* shows how the overall synaptic conductance ($G_{syn}$; after subtraction of rest conductance) varies during the first 100 ms of the synaptic response, as plotted against the estimated reversal potential ($V_{rev}$). The overall shape of this relationship for visual (blue dots), electrosensory (red dots) and bimodal (black dots) is very similar on a cell-to-cell basis. The few points with a reversal potential near zero represent the initial portion of the response when excitation dominates. However, the peak conductance is at a reversal potential near −48 to −50 mV. The peak conductance

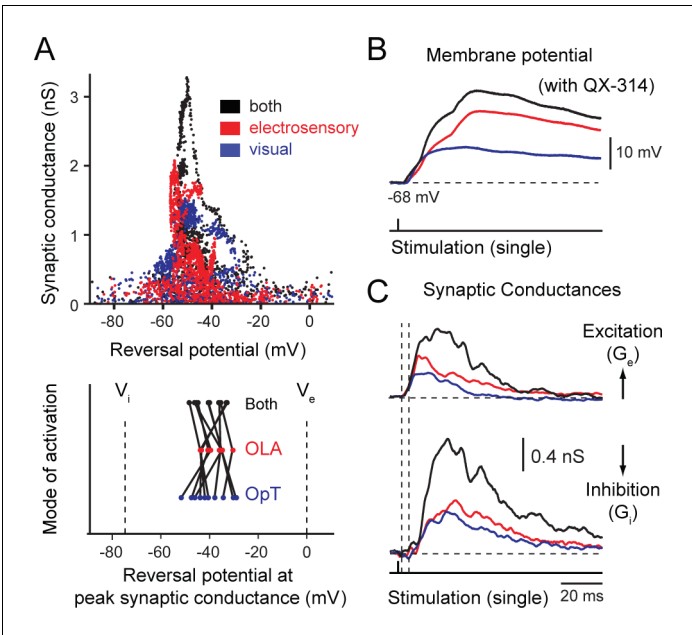

**Figure 6.** Dynamics between excitation and inhibition during unimodal and bimodal integration. (**A**) (Top) Phase plot showing the estimated change in synaptic conductance of an output neuron (shown in **B**) plotted against the reversal potential during OpT (blue), OLA (red) and bimodal (black) afferent stimulation. Conductances were estimated at each time point for the interval ranging from 10 ms before the impulse until 100 ms after stimulation onset. (Bottom) Line graph highlighting the paired differences between reversal potential across cells. Lines link measurements across conditions performed on the same neuron. On average, there is no statistical difference between unimodal and bimodal integration (*n* = 10). (**B**) A representative example of the synaptic responses evoked by single impulses of the OpT, OLA and both from an output neuron resting at −65 mV. (**C**) Time course of the estimated underlying synaptic conductances for excitation and inhibition is shown (average data from *n* = 10). Drop lines show the onsets of excitation and inhibition (which shows a delay with respect to excitation of ~5–10 ms).

values for unimodal inputs were visual: 1.6 ± 0.2 nS, electrosensory: 2.1 ± 0.3 nS; *n* = 10, while an increase of ~150–200% occurred during bimodal stimulation (bimodal: 3.3 ± 0.6 nS). Strikingly, these values were observed for the corresponding reversal potentials, which were −48.2 ± 2.7 mV for visual inputs, −50.4 ± 2.2 mV for electrosensory and −48.2 ± 2.4 mV for combined inputs. These values are near the action potential threshold (−43.7 ± 1.0 mV, *n* = 10; *Figure 6A*, bottom). When the total synaptic conductance is at a maximum, the reversal potential target values that are relatively invariant, despite the additional excitatory current when switching from unimodal to bimodal input.

The latency between the onset of excitation and inhibition is time-locked (*Figure 6C*) for both unimodal and bimodal inputs, as well as the latency between their peaks. *Figure 6B* shows a representative example of the time course of an excitatory response pattern belonging to an output neuron (in current clamp) in response to the underlying excitatory ($G_e$) and inhibitory synaptic conductances ($G_i$; population average of *n* = 10, lower traces; *see Figure 6C*). Visual, electrosensory and bimodal peak values of $G_e$ and $G_i$ were found to be visual: $G_e$ = 1.11 ± 0.09 nS, $G_i$ = 0.89 ± 0.08 nS; electrosensory: $G_e$ = 1.08 ± 0.07 nS, $G_i$ = 1.13 ± 0.17 nS; bimodal: $G_e$ = 1.96 ± 0.53 nS, $G_i$ = 1.81 ± 0.31 nS; *n* = 10). The latency between the onset of $G_i$ and $G_e$ was between 5–10 ms (*n* = 10; see arrows in *Figure 6C*, and *Kardamakis et al. [2015]*), thus, providing a temporal window of opportunity for output neurons to integrate independent excitatory inputs from each sensory modality. This time lag is sufficient for transient stimuli to efficiently summate inputs towards spike threshold before the onset of the inhibition. The quenching effect of this feedforward inhibition can account for the mismatch between the evoked EPSP amplitudes to their underlying excitatory conductances during bimodal stimulation, i.e. the subadditive effect of combining both sensory inputs. Furthermore, the summating effect observed in the synaptic current measurements obtained in voltage clamp

(*Figure 5A*) is also reflected in $G_e$ and $G_i$ (*Figure 6C*). This close agreement suggests that non-linearities in the form of strong voltage-dependent conductances or shunting inhibition are not likely to play a significant role in the gain regulation of responses in output neurons during multisensory integration.

## Temporal offset leads to amplitude attenuation

To test how the timing of sensory inputs affects the amplitude of their excitatory responses, we systematically altered the stimulation onset of either sensory input from 0 to 50 ms (*Figure 7A,B*). The maximal response was obtained when the onset of both sensory-evoked EPSPs was aligned so that excitation would be in-phase, i.e. latency of 0 ms (*Figure 7A*, left & 6B, black trace), whereas amplitude reduction occurred when afferent stimulation onsets were temporally misaligned using an offset from 5 to 50 ms. For instance, when incoming afferent excitation coincided with a prior afferent-triggered inhibitory event, as in the case of a 5–10 ms delay (*Figure 7B*, orange trace), an attenuation of the resultant EPSP was observed. Temporal summation was greater for sensory inputs that were separated further apart to the peak of any preceding inhibition, i.e in the decay phase (typically for latencies > 20 ms; *Figure 7B*, green trace). To quantify this effect, we selectively stimulated each sensory afferent pathway (with 10 pulses at 10 Hz; *Figure 7A*) and normalized the combined amplitudes to the first temporally-aligned EPSP (*Figure 7C*). The largest combined responses were generated when both stimuli were aligned. A maximal attenuation of 15% was achieved with a 5–10 ms latency (see also *Figure 7B*, orange trace), matching the maximum peak of inhibition, with a subsequent increase with successively longer delays greater than 10 ms (*n = 10*; *Figure 7C*). Thus, the sequence of sensory modality inputs did not have a differential impact on the attenuation level. Taken together, the timing of sensory-evoked inhibitory events regulates the responsiveness of output neurons.

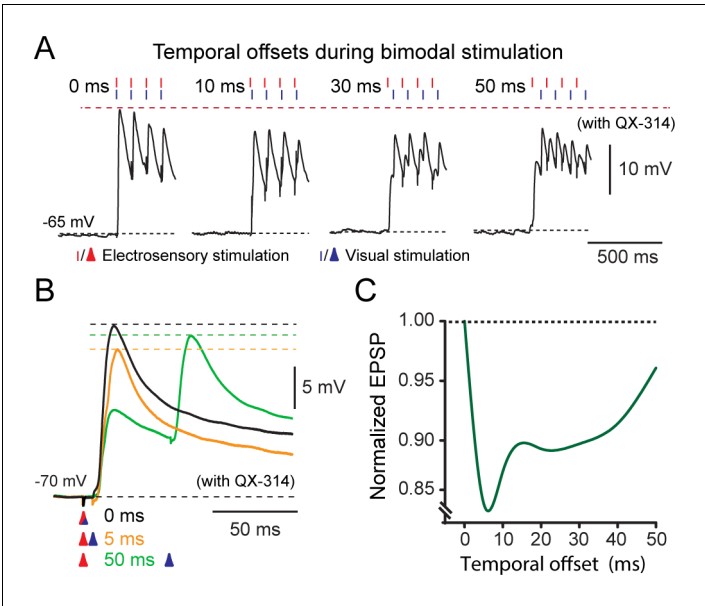

**Figure 7.** Temporal effects on bimodal integration. (**A**) The response profile of a representative output neuron during repetitive bimodal stimulation (4 pulses at 10 Hz) of OpT and OLA. Traces from left to right show the excitatory postsynaptic potentials with offsets ranging from 0 (temporal alignment), 10, 30 and 50 ms. (**B**) Recordings of EPSPs evoked after variable activation of OpT and OLA with temporal offsets of: 0 ms (aligned; black trace), 5 ms (orange trace) and 50 ms (green trace). (**C**) Curve fit illustrating the variation of the combined excitatory postsynaptic potential amplitude against the temporal offset between the two sensory modalities. The graph was normalized to the maximal EPSP amplitude, which always occurred when inputs were temporally aligned (i.e., 0 ms). Average shown from *n = 7*.

## Bimodal sensory inputs also drive inhibitory interneurons

The key finding that bimodal inputs to the optic tectum can trigger inhibitory responses raised the question of the location of the cells of origin. By removing the diencephalon and ventral parts of the midbrain (*see Figure 3A*), we excluded potential extrinsic sources of inhibition, thus, enabling us to limit the source of inhibition to within the tectum. As seen in *Figure 8A-i,ii*, GABAergic interneurons are strategically located within the stratum opticum - at the interface between the superficial and intermediate layers (*Isa et al., 1998*; *Del Bene et al., 2010*; *Kardamakis et al., 2015*). Previously, we have shown that microstimulation of this population of tectal interneurons elicits monosynaptic inhibition onto output neurons in the presence of glutamate blockade (2 mM Kynurenic acid) to eliminate excitatory transmission (*see Figure 2* in *Kardamakis et al., 2015*).

To analyze the activation of these cells by visual and electrosensory inputs, we performed whole-cell recordings while microstimulating both sensory pathways. EPSPs were evoked (*Figure 8B*) after a train of stimuli (at 10 Hz) in cells held at −65 mV in current-clamp, both after electrosensory (*Figure 8B–i*) and visual stimulation (*Figure 8B-ii*), showing that the same pool of interneurons is activated by both sensory modalities. Remarkably, sensory stimulation not only evoked excitation (EPSPs, visual: 8.6 ± 1.5 mV, electrosensory: 5.6 ± 0.8 mV, n = 8), but also a prominent inhibition (when holding at −45 mV; IPSPs, visual: 9.8 ± 1.6 mV, electrosensory: 9.8 ± 1.7 mV, n = 8). In addition, these interneurons have a strikingly different morphology (*Figure 8C*) compared to that of output neurons (*Figure 1A*). *Figure 8C* illustrates six interneurons that have broad dendritic arbor extending in all planes (over around 200 μm), and also bridge the layers of retinal and electroreceptive fiber termination (as shown in *Figure 1A*). In half of the neurons that were stained (3/6), it was possible to also see long range axonal projections, as required for the lateral inhibition.

To test if superficial layer interneurons are activated in a spatiotopic manner, like deep layer output cells, we used a preparation exposing the optic tectum to allow patch-clamp recordings from the superficial layer while keeping the retina and the optic nerve intact, so that local stimulation was

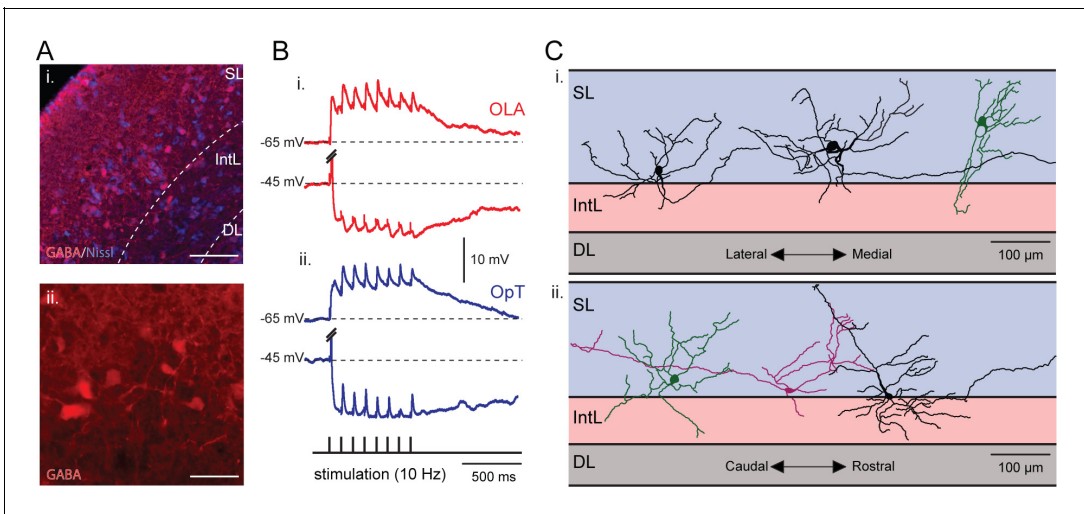

**Figure 8.** Visual and electrosensory inputs activate the same set of interneurons. (**A**) *Inset* i: A neural population located at the interface between the superficial and intermediate layers express GABA. These cells can be seen in more detail in *inset* ii. Scale bars: *Inset* i, 150 μm; *Inset* ii, 50 μm. (**B**) An example of a tectal interneuron that receives excitatory inputs from both sensory modalities (visual and electrosensory) along with triggered tectal inhibitory inputs. Here, EPSPs (hold at −65 mV) and IPSPs (hold at −45 mV) are shown in response to repetitive stimulation of the OLA (red, bottom traces) and OpT (blue, top traces). The action potentials evoked by the initial impulse when holding at −45 mV are truncated. The morphology of this cell is shown in *Figure 8—figure supplement 1*. (**C**) Six reconstructed morphologies of interneurons that were stained with neurobiotin, in the transversal (*inset* i) and sagittal (*inset* ii) dimensions. Scale bars: 100 μm.

The following figure supplement is available for figure 8:

**Figure supplement 1.** A representative example of the morphology of a tectal interneuron that was filled with neurobiotin while performing the whole-cell recordings shown in *Figure 8B*.

possible (*Figure 9A*, top; *see Kardamakis et al., 2015* for experimental procedure). By dividing the retina into four quadrants, we determined the one that gave rise to on-receptive field responses by using cell-attached (*Figure 9A*, bottom) and whole-cell recordings (*Figure 9B*) from tectal interneurons located in the stratum opticum. As with tectal output neurons, action potentials time-locked to the stimulus were recorded during on-receptive field activation, but disappeared usually after the second pulse (in 5 out of 7 neurons) due to synaptic depression. A representative example of the interplay between visually-evoked synaptic excitation and inhibition in tectal interneurons can be seen when entering whole-cell configuration (*Figure 9B*) with subthreshold EPSPs (red trace)

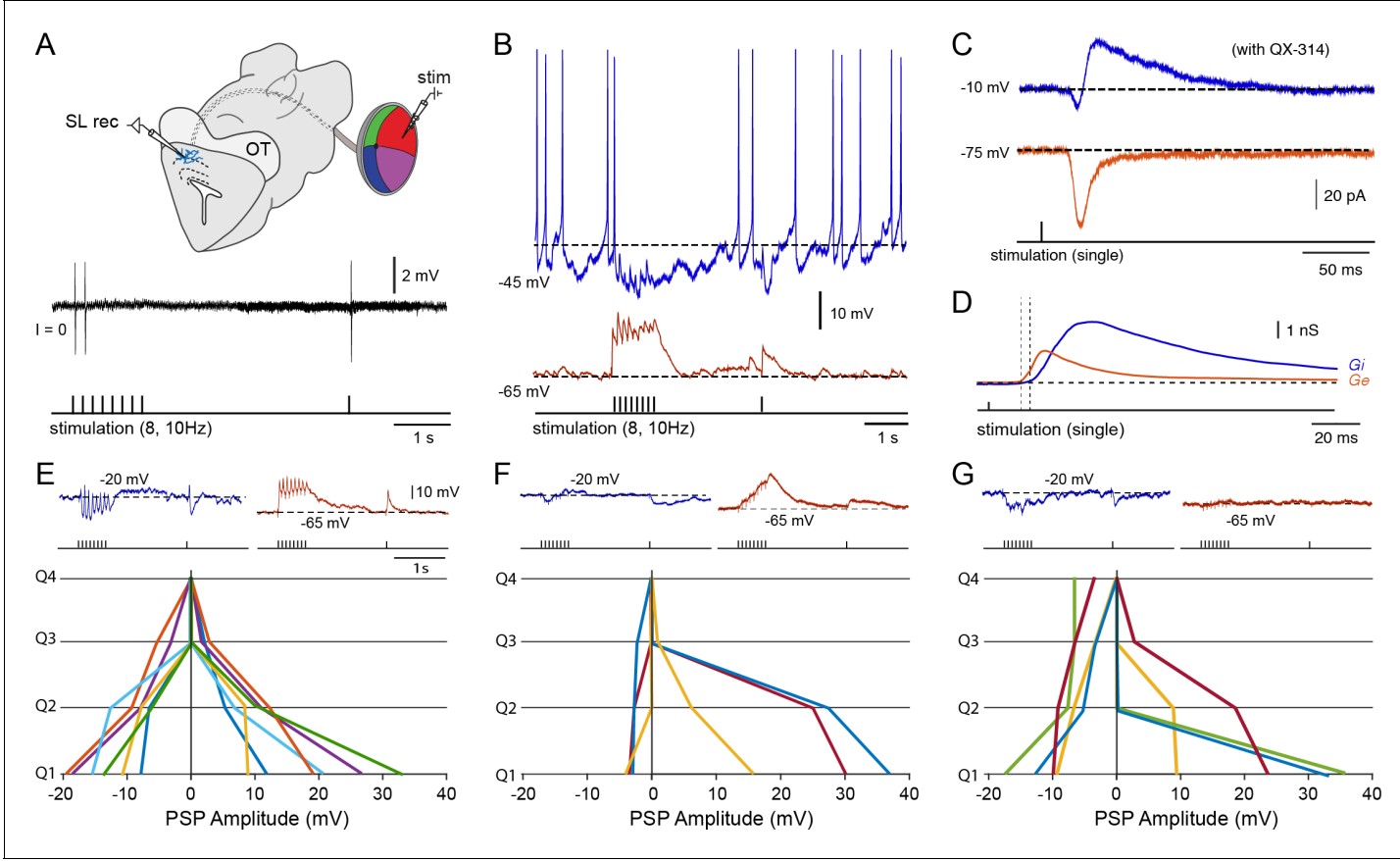

**Figure 9.** Tectal interneurons receive diverse forms of local inhibition. (**A**) (Top) Schematic of the in vitro eye-brain preparation used to locally stimulate regions of the retina (divided in four quadrants) while performing whole-cell recordings in the superficial layer interneurons located in the stratum opticum. (Bottom) Cell-attached recording of a tectal interneuron in response to electrical microstimulation of the on-receptive field in the retina (stimulation train of 8 pulses at 10 Hz with a recovery pulse; threshold ~50 µA, n = 7). As with tectal output neurons (*see Figure 3B*), one time-locked action potential can be observed in response to an impulse delivered to the preferred retinal quadrant, with depression quickly following the first or second impulse. (**B**) Whole-cell current clamp recording of an interneuron held at the reversal for chloride, −65 mV, and near threshold ~45 mV using positive somatic current injection. (**C**) Whole-cell voltage recordings of an interneuron clamped at −65 and 0 mV using QX-314 in the pipette solution to block action potential generation (average of 10 sweeps). (**D**) Decomposition of the underlying excitatory ($G_e$; red) and inhibitory synaptic conductances ($G_i$; blue) arising from electrical microstimulation of the preferred quadrant. Drop lines indicate the onsets of excitation and inhibition. (**E–G**) Quantification of the maximal excitatory (depolarization from −65 mV) and maximal inhibitory amplitudes (hyperpolarization from −20 mV) in current clamp arising from selective activation of each of the four retinal quadrants (dorsal, ventral, anterior and posterior) using a train of 8 pulses at 10 Hz (n = 10). However, to avoid dependencies on the particular stimulated retinal area, we sorted the quadrants that give rise to the strongest excitatory component in descending order (Q1 being the largest EPSP). Lines with matching colors on each side of the y-axis represent the postsynaptic potential amplitudes obtained from individual recorded interneurons (positive values represent the total EPSP size and negative values represent the total IPSP size). We determined three subtypes of interneurons on the basis of their inhibitory response patterns. In **E**, neurons displayed inhibitory responses only when excitation was present, which usually was broad and spread into neighboring quadrants (top: typical traces obtained from Q1). In **F**, interneurons received weak synaptic inhibition (top: typical build-up excitation obtained Q1 & Q2). In **G**, interneurons received local excitation and widespread inhibition arising from quadrants that did not display excitation (top: typical off-receptive inhibition arising from Q3 & Q4).

accompanied by inhibitory postsynaptic potentials visible when depolarizing near threshold (~ −45 mV) with positive somatic current injections (blue trace). Voltage clamp recordings reveal the sequence of the inward excitatory (red) and outward inhibitory synaptic currents (blue) in response to the on-receptive field quadrant stimulation of the retina (*Figure 9C*). This is further decomposed into the underlying synaptic conductance in *Figure 9D*, where temporal lag of inhibition with respect to excitation is reflected.

To identify the synaptic inputs that arise from the four retinal quadrants (Q1, Q2, Q3 & Q4), we measured the maximal amount of depolarization from −65 mV and hyperpolarization from −20 mV that was generated with a stimulation train of 8 pulses at 10 Hz. All interneurons that were recorded received excitation from usually one main quadrant and in some cases (6 out of 13) they also received from neighboring ones. Their inhibitory responses, however, displayed a range of variability by which we subdivided into three provisional groups: (a) local excitation and local inhibition (*Figure 9E*), (b) local excitation and no inhibition (*Figure 9F*) and (c) local excitation and global inhibition (*Figure 9G*). We sorted the responses that generated the largest PSPs by quadrant starting from Q1 through Q4 (Q1 being the strongest). Despite the variable nature of visually-evoked synaptic inhibition onto tectal interneurons, the spatiotopic arrangement of their excitatory components is critical for the feedforward inhibition onto tectal output neurons.

## Discussion

This study relies on the fact that gaze controlling output neurons in the deep layer of the lamprey optic tectum (a) receive visual input from the retina and electroreceptive input from octavolateral afferents, (b) are excitatory and target the reticulospinal system within the brainstem that control gaze movements (*Video 1*; *see* also *Saitoh et al., 2007*) and (c) are easily identified and accessed for monitoring their activity. Upon entering the optic tectum, afferents arising from both the retina and the octavolateral area, contact monosynaptically the same output neurons, as well as inhibitory interneurons (GABAergic cells; *Figure 10*), which in turn inhibit output neurons across the tectal map of space. This disynaptic circuit is able to integrate excitation from different sensory stimuli and feedforward inhibition via interneurons, with spatial and temporal correspondence, to control gaze movements.

### The role of tectal inhibition

The tectal GABAergic system regulates the incoming excitatory sensory flow to deep layer output neurons and controls their response profile to stimuli of different modalities. It is responsible for the suppression of stimuli with spatial and temporal offsets.

Recently, we have shown that on-receptive field retinal stimuli result in local excitation followed by feedforward inhibition, while off-field retinal stimulation yields only lateral inhibition, which is widespread (*Kardamakis et al., 2015*). This inhibition is carried by short- and long-range connections spanning rostrocaudally and mediolaterally across the optic tectum (*Phongphanphanee et al., 2014*; *Kardamakis et al., 2015*; *Figure 8*). While this local inhibition can act independently for generating stimulus selection, it has also been suggested that exogenous sources are required for generating global inhibition via GABAergic neurons in the isthmic nucleus (*Mysore and Knudsen, 2013*). Here, we show that inhibition hardwired in the optic tectum via interneurons, that are activated from both retinal and

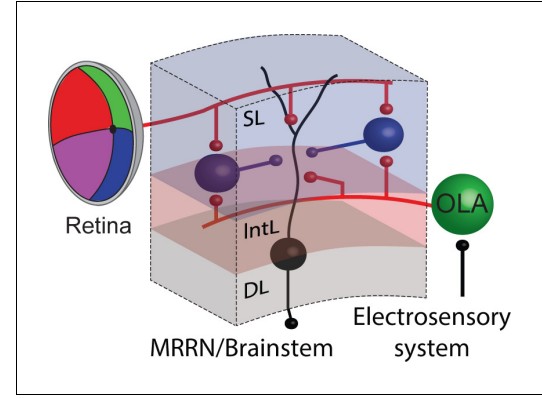

**Figure 10.** Summarizing circuit. Tectal output cells (black) in the deep layer receive a combination of visual and electrosensory synaptic input arising from the retina and the octavolateral area, respectively. Tectal interneurons in the superficial layer (putatively inhibitory and therefore shown in blue) also receive similar sensory inputs from both the visual and electrosensory system, and once activated in turn inhibit their output neurons. Abbreviations: SL, superficial layer; IntL, intermediate layer; DL, deep layer; OLA, octavolateral area; MRRN, middle rhombencephalic reticular nucleus.

electroreceptive afferents in the same way as the output neurons, is essential for multisensory stimulus selection. Notably, the two types of sensory inputs not only evoke excitation in tectal interneurons, but also trigger recurrent inhibition.

Our results suggest that spatiotopic organization in the optic tectum is columnar, with output neurons and interneurons receiving retinal excitation from the same quadrant, as suggested by their morphological features and the fact that the excitation is retinotopically arranged. However, the detailed arrangement of columnar structure and function has yet to be revealed. Sensory inputs provide monosynaptic drive to the deep layer output cells and to superficial layer interneurons. The latter, in turn, provide local and long-range inhibitory projections, therefore allowing for spatial sensory discrimination. This role of inhibition is observed in our extracellular recordings. Consistent with bimodal suppression in mammals (*Meredith and Stein, 1996*; *Kadunce et al., 1997*), responses in on-field regions of tectum to electrosensory stimulation undergo a drastic reduction when delivering visual stimuli to other receptive fields of the tectal map. In the present study, we could not elicit the same sort of reduction in tectal areas responsive for visual stimulation when applying off-field electrosensory stimuli. One possibility is that electrosensory maps in the lamprey tectum are not as refined as the visual ones, and therefore inhibition is less effective suppressing off-regions. In this sense, it has been shown that electrosensory receptive fields in elasmobranchs tectum are much larger than the visual ones (*Bodznick, 1990*). We cannot exclude either that the way we perform electrosensory stimulation is not as local as the visual one, making it less effective to evoke a similar reduction. Consistent with our previous findings, these data suggest that inhibitory connections embedded within the tectal circuit can account for how normally effective stimuli can be attenuated by the presence of bimodal stimuli when they are spatial and/or temporally misaligned.

## Response enhancement with multisensory integration

Despite the powerful action of the GABAergic system, it *does* allow for response enhancement during bimodal integration. The output neurons yield responses of matching amplitudes to both types of sensory inputs, without any apparent domination by a particular modality. Two overlapping (in space and time) stimuli from these two modalities will produce a stronger response, than when only one modality is present. As a rule, two sensory modalities that provide stimuli spatially and temporally aligned are most likely related to the same external event. In this manner, the response enhancement caused by the integration of two senses gives rise to higher event detection reliability. When this occurs, bimodal inputs are able to increase the amount of depolarization in output neurons and increase the probability of evoking an action potential, and therefore, to activate their downstream targets and elicit a motor response.

Tectal inhibition will continuously reset active areas to enable the constant monitoring of information flow from the environment. Without this inhibition, response enhancement will not be possible since tectal output cells rapidly saturate most likely due to a ceiling effect, making neurons incapable of further responding and thereby losing their response sensitivity, and hence, event detection reliability. Tectal inhibitory action can be interpreted as a gain modulatory mechanism whose effects are well captured by the conductance-based analysis of output neurons, and visible throughout a wide range of membrane potentials (i.e., varying from below to above spike threshold). Local circuit dynamics triggered by sensory inputs will allow for depolarizing synaptic events to reach the reversal potential of the synaptic response. We show that this value is invariant during unimodal or bimodal stimulation implying that a strong unimodal stimulus can bring the membrane potential to action potential threshold, as well as two weaker bimodal stimuli. When bimodal stimuli overlap, they will reinforce each other and generate larger depolarizing currents that will increase the likelihood that the output neurons reach spike threshold before inhibition fully develops. This enhancement most likely arises from sensory inputs that target the different dendritic compartments of output neurons (i.e., superficial vs intermediate layer), that were previously inactive during unimodal stimulation.

## Top-down regulation of the optic tectum from the forebrain

Although our findings imply that tectum can perform stimulus selection without the external control from any other brain region, activity in the deep layer output neurons can be strongly influenced by forebrain structures such as the cortex and the basal ganglia. As in mammals, tectal output neurons in lamprey also receive the combination of inhibitory and excitatory input from the forebrain. The

GABAergic output neurons of the basal ganglia output nuclei provide tonic inhibition at rest in both lamprey and primates, and can trigger movements by way of disinhibition of the tectal output neurons (*Wurtz and Hikosaka, 1986*; *Stephenson-Jones et al., 2012*; *Kim and Hikosaka, 2015*). Pallium (evolutionary forerunner of the mammalian cortex) projects to the lamprey optic tectum and provides monosynaptic excitation to the output neurons in the deep layer, and its stimulation is able to elicit eye-head gaze movements (*Ocaña et al., 2015*). In the forebrain control of tectum, it would seem likely that the basal ganglia disinhibition will be complementary to the excitation conveyed from pallium/cortex, and in both cases they target the soma level of the output neurons rather than the dendrites.

Extracellular recordings in cats and primates have shown a supra-linear enhancement of deep layer collicular activity in response to overlapping weak multimodal stimuli (*Meredith and Stein, 1983*; *Wallace and Stein, 1994*; *Wallace et al., 1996*; *Stein, 2005*). This amplification is thought not to arise from the intrinsic collicular circuitry, but from extrinsic cortico-collicular afferents from the anterior ectosylvian sulcus. When this area is inactivated the amplification vanishes (*Wallace and Stein, 1994*; *Jiang et al., 2001*; *Alvarado et al., 2009*). One mechanism would be to bypass tectal inhibition by selectively increasing excitation onto the output neurons, which are targeted by monosynaptic projections from cortex/pallium in both mammals and lamprey. Another potential mechanism that would introduce such non-linearities could be a selective decrease of inhibition by briefly 'switching off' the tectal GABAergic system. The variable classes of tectal interneurons may also provide an additional flexibility to manipulate sensorimotor processing within the optic tectum.

## Conclusion

The basic anatomical and functional organization of the optic tectum across vertebrates is highly conserved, from the earliest group of vertebrates that has evolved - the lamprey (*Nieuwenhuys and Nicholson, 1998*; *Saitoh et al., 2007*; *Jones et al., 2009*; *Asteriti et al., 2015*; *Kardamakis et al., 2015*), despite the particular sensory modalities that the different species depend on. The lamprey optic tectum consists of glutamatergic and GABAergic neurons activated by unimodal and bimodal inputs. The output neurons found in the deep layer are excitatory and project to the brainstem, and control gaze movements (*Saitoh et al., 2007*). In lamprey, visual and electrosensory stimuli evoke direct excitation quickly followed by recruited inhibition onto these output neurons. These synaptic responses are amplified when spatiotemporally aligned bimodal inputs are present, thus, leading to a response enhancement. We have shown here that sensory-evoked inhibition scales the neural activity of output cells to allow for a robust spatiotemporal integration, capable of integrating the enormous amount of incoming sensory inputs during natural conditions. Local inhibition may be critical for spatiotemporal processing not only in lamprey but also in other vertebrates.

## Materials and methods

### Animals

Experiments were performed on 47 adult river lampreys (*Lampetra fluviatilis*). During the investigation, every effort was made to minimize suffering and to reduce the number of animals used, in accordance with the Guide for the Care and Use of *Institute of Laboratory Animal Research, National Research Council (1996)*.

### CLARITY-optimized light-sheet microscopy (COLM)

After injecting the tracer Neurobiotin (20%, see above), brains were dissected and incubated with CLARITY monomer solution containing 1% acrylamide, 0.0125% bis-acrylamide, and 4% PFA and then polymerized at 37°C for 8 hr. The brains were passively cleared in SDS Borate Buffer (pH 8.5) at 37°C for 2 weeks and then equilibrated in 0.2 M Borate Buffer (pH 7.6) containing 0.1% Triton. The brains were incubated with Streptavidin Alexa Fluor 488 (1:100) in the same buffer at 37°C for 4 days. Sequential washing followed prior to equilibrating to a final concentration of 65% glycerol containing Anti-fade for imaging. Once mounted they were imaged using COLM methods described in *Tomer et al. (2014)*.

## Double-tracing

The animals ($n = 3$) were deeply anesthetized with tricaine methane sulfonate (MS-222) (100 mg/L; Sigma) diluted in fresh water. During the surgery and the injections, the entire animal was submerged in ice-cooled artificial cerebrospinal fluid (aCSF) solution containing the following (in mM): 125 NaCl, 2.5 KCl, 2 CaCl$_2$, 1 MgCl$_2$, 10 glucose, and 25 NaHCO$_3$, saturated with 95% (vol/vol) O$_2$/5% CO$_2$. An incision was performed directly above the octavolateral area to expose the brain. All injections were made with glass micropipettes (borosilicate; o.d. = 1.5 mm, i.d. = 1.17 mm; Hilgenberg) with a tip diameter of 10–20 µm. The micropipettes were fixed to a holder attached to an air supply and a Narishige micromanipulator. Fifty to 200 nL of Alexa Fluor 488-dextran (10 kDa; 12% (wt/vol) in saline; Molecular Probes) were pressure injected unilaterally into the octavolateral area. Subsequently, an incision was performed in the primary spectacle, the lens was removed to expose the retina, and Neurobiotin [20% (wt/vol) in aCSF containing Fast Green to aid visualization of the injected tracer; Vector Laboratories], was injected in the central retina, ipsilateral to the octavolateral area injection. Following injections, the dorsal skin and the spectacle were sutured, and the animal was returned to its aquarium for 48–72 hr to allow transport of the tracers. The brains were then dissected out and fixed in 4% formaldehyde and 14% saturated picric acid in 0.1 M phosphate buffer (PB), pH 7.4, for 12–24 h, and then cryoprotected in 20% (wt/vol) sucrose in PB for 3–12 hr. 20-µm-thick transverse sections were made using a cryostat, collected on gelatin-coated slides, and stored at −20°C until further processing. For detection of Neurobiotin, Cy2 conjugated streptavidin (1:1000; Jackson ImmunoResearch, PA) and a deep red Nissl stain (1:1000; Molecular Probes) were diluted in 1% bovine serum albumin (BSA), 0.3% Triton X-100 in 0.1 M PB. All sections were mounted with glycerol containing 2.5% diazabicyclooctane (Sigma-Aldrich). To label brainstem projecting neurons for patch-clamp experiments, tetramethylrhodamine-dextran (3 kDa; 12% in saline; Molecular Probes) was pressure injected unilaterally into MRRN (Middle Rhombencephalic Reticular Nucleus) in the brainstem.

## Morphology

Prelabeled brainstem-projecting cells and superficial layer interneurons were intracellularly injected with 0.3–0.5% Neurobiotin (Vector Laboratories) during patch-clamp recordings. Brain slices were fixed overnight in 4% formaldehyde and 14% picric acid in 0.1 M PB. Following a thorough rinse in PBS, the slices were incubated in streptavidin-Cy2 (1:1000, Jackson ImmunoResearch) in 0.3% Triton X-100 and 1% BSA in 0.1 M PB for two hours at room temperature. The slices were then rinsed in 0.01 M phosphate buffered saline (PBS) and mounted in glycerol containing 2.5% diazabicyclooctane (DABCO; Sigma). Labeled cells were analyzed by either confocal or conventional fluorescence microscopy.

## Extracellular recordings

Local field potentials (LFPs) were recorded using tungsten microelectrodes (~1–5 MΩ, a 4-channel MA 102 amplifier and a MA 103 preamplifier (Elektroniklabor, Zoologie, University of Cologne), and digitized at 20 kHz using pClamp (version 9.2) software. For natural sensory stimulation, the head was transected from animals deeply anesthetized with MS-222 (100 mg/L; Sigma) and the dorsal skin and cartilage were removed to expose the brain. The skin with the electrosensory receptive organs in the rostrolateral part of the head was kept, and the muscles in the ventral part were removed. To avoid movements that could destabilize the preparation, the neuromuscular blocker α-bungarotoxin (12.5 µM, Sigma) was locally injected into the muscles that could not be removed. Visual stimulation was performed with brief flashes of light (500 ms duration), using a 100 µm diameter-thick optic fiber connected to a standard LED light source. The optic fiber was connected to a borosilicate glass pipette painted black with nail polisher to avoid light spread, so that the light spot diameter was ~50 µm. The pipette was attached to a Narishige micromanipulator, and placed ~2 cm distance from the retina, and once the receptive field in the optic tectum (OT) was located by recording LFPs, an electric field was spatially aligned with the visual stimulus. To generate it, two copper wires (used respectively as negative and positive poles) were connected to a stimulus isolation unit (MI401; Zoological Institute, University of Cologne), and submerged in the aCSF at a distance of ~5–10 cm from the preparation, keeping a small distance between poles to ensure a local stimulus (~1 cm).

Electrosensory stimulation was presented as brief pulses (30 ms in duration) with intensities between 10–100 µA.

For electric stimulation of the retina and the anterior line nerve (ALLN), the procedure was the same, but the cartilage of the otic capsule was removed in order to expose the ALLN. The stimulation was performed by using borosilicate glass microcapillaries connected to a stimulus isolation unit (MI401; Zoological Institute, University of Cologne). The stimulation intensity was set to the threshold strength (typically 10–100 µA) necessary to evoke LFPs with a 50 µs duration stimulus, and consecutive increasing stimulation durations of 100, 200, 300, 400, 500, 700 and 1000 µs were then applied. All experiments were performed in darkness to avoid interfering visual stimuli.

## Whole-cell recordings

A novel preparation was used to perform whole-cell current-clamp recordings, slicing a thick brain section exposing the deep layer of the optic tectum, keeping the retinal and octavolateral afferent tracts intact (see *Figure 2A*). To expose the different layers of the optic tectum by sectioning, the entire preparation was first embedded in agar (4% dissolved in aCSF; Fluka). The agar block containing the brain was then cut in an oblique angle and glued to a metal plate, quickly transferred to ice-cold aCSF and sagittal-oblique slices were cut using a vibrating microtome (Microm HM 650V; Thermo Scientific) until the deep layer of tectum as well as the retinal and octavolateral tracts were exposed. The agar block was then mounted in a submerged recording chamber.

Whole-cell voltage and current-clamp recordings were performed with patch pipettes made from borosilicate glass (Hilgenberg) using a vertical puller (Model PP-830; Narishige). The resistance of recording pipettes was 7–10 MΩ when filled with intracellular solution of the following composition (in mM): 130 potassium gluconate, 5 KCl, 10 phosphocreatine disodium salt, 10 HEPES, 4 Mg-ATP, 0.3 Na-GTP; (osmolarity 265–275 mOsmol). The electrode solution also included in some cases 3 mM of triethylammonium bromide (QX-314; Sigma) to block action potentials. Bridge balance and pipette-capacitance compensation were adjusted for using a MultiClamp 700B patch amplifier and Digidata 1322 analog-to-digital converter under software control 'PClamp' (Molecular Devices). Perfusion of the preparation was performed with aCSF at 6–8°.

Stimulation of the octavolateral and retinal afferents was performed with borosilicate glass microcapillaries (the same as for patch recordings), connected to a stimulus isolation unit (MI401; Zoological Institute, University of Cologne). The stimulation intensity was set to one to two times the threshold strength (typically 10–100 µA) to evoke PSPs. To investigate the short-term dynamics of synaptic transmission, a stimulus train of ten pulses at 10 Hz was used (*Ericsson et al., 2013*).

## Temporal offset analysis

To examine the effect on the bimodal integration of the temporal overlap between visual and electrosensory modalities, we performed patch-clamp recordings using asynchronous visual and electrosensory stimulations, delaying one stimulus respect to the other in steps of 5, 10, 20, 30, 40 and 50 ms. To temporally align visual and electrosensory responses, the EPSPs evoked by the first pulse of each sensory modality were analyzed online for every single cell to calculate the difference between the onsets. This difference was then used to adjust the timing for the different stimuli steps, which were programmed using a Master-8 programmable pulse generator (AMP Instruments LTD).

## Drug applications

During extracellular recordings, the GABA$_A$-receptor antagonist (Gabazine; 10 µM; Tocris) was locally applied in the deep layer of the optic tectum by pressure injection through a micropipette fixed to a holder, which was attached to an air supply and a Narishige micromanipulator. For patch-clamp recordings, gabazine was bath applied.

## Immunohistochemistry

For immunohistochemical detection of GABA, sections were incubated overnight with a mouse monoclonal anti-GABA antibody (1:5000; mAb 3A12; kindly donated by Dr. Peter Streit, Brain Research Institute, University of Zürich, Zürich, Switzerland). The sections were subsequently incubated for 2 hr at room temperature with a Cy3-conjugated donkey anti-mouse IgG (1:500; Jackson ImmunoResearch) and, in those cases in which the cells were filled with neurobiotin, also with Cy2-

conjugated streptavidin (1:1000; Jackson ImmunoResearch). All primary and secondary antibodies were diluted in 1% BSA and 0.3% Triton-X 100 in 0.1 M PB.

## Image analysis

Photomicrographs were taken with an Olympus XM10 digital camera mounted on an Olympus BX51 fluorescence microscope (Olympus Sweden). Illustrations were prepared in Adobe Illustrator and Adobe Photoshop CS4. Images were only adjusted for brightness and contrast. Confocal Z-stacks of optical sections were obtained using a Zeiss Laser scanning microscope 510, and the projection images were processed using the Zeiss LSM software, ImageJ and Adobe Photoshop CS4.

## Data analysis

For all electrophysiological recordings, data analysis was performed using custom written functions in Matlab (*see Source code 1*). For extracellular recordings, the integral under the curves were compared after fully rectifying the signals using trapezoidal numerical integration ('*trapz*' function). For patch-clamp recordings, subsequent PSPs often started on the decay phase of previous responses, so that to extract correct amplitudes the synaptic decay was either fitted by an exponential curve and subtracted or manually subtracted.

For the estimation of the synaptic conductances, we used the direct extraction of the excitatory and inhibitory conductance based on solving the conductance model equation applied to the current-clamp data as explained in p. 328–329 in *Monier et al. (2008)*. In all recordings, these estimates were accurate because (i) the synaptic responses were composed primarily of ionotropic glutamatergic synapses (primarily AMPA, see also *Kardamakis et al., 2015*) and chloride-mediated GABAergic (GABA$_A$) synaptic inputs, (ii) their reversal potentials were experimentally determined and found to be at 0 and −75 mV, respectively, and (iii) due to the linearity domain of their voltage-current relationships and similar time constants.

For statistical analysis, we used two-sample unpaired and paired *t*-tests in Matlab. Throughout the figures, sample statistics are expressed as Means ± SEMs (SEM; standard error), unless specified otherwise.

## Acknowledgements

The experiments were funded by grants from the Swedish Research Council (both Health and Science and Technology), the EU and the Karolinska Institute (Strategic Research). Comments on the manuscript and advice throughout the experiments from Drs Brita Robertson, Abdel El Manira and Peter Wallén are gratefully acknowledged. We are also grateful for being able to use the light-sheet microscopy developed by Dr. Raju Tomer, Professor Ole Kiehn and Dr. Peter Löw in the Department of Neuroscience, and in particular to the generosity of Dr. Carmelo Bellardita who imaged the lamprey brains.

## Additional information

### Funding

| Funder | Grant reference number | Author |
| --- | --- | --- |
| Vetenskapsrådet | VR-M-K2013-62X-03026 | Sten Grillner |
| Seventh Framework Programme | FP7/2007-2013 under grant agreement no 604102 (HBP) | Sten Grillner |
| Vetenskapsrådet | VR-NT-621-2013-4613 | Sten Grillner |
| Karolinska Institutet | Strategic grant | Sten Grillner |

The funders had no role in study design, data collection and interpretation, or the decision to submit the work for publication.

## Author contributions
AAK, JP, Wrote the paper, Performed the experiments and analyzed data, Designed research; SG, Wrote the paper, Designed research

## Author ORCIDs
Sten Grillner, http://orcid.org/0000-0002-8951-3691

## Ethics
Animal experimentation: Experiments were performed on adult river lampreys (Lampetra fluviatilis). The experimental procedures were approved by the local ethics committee (Stockholms Norra Djurfrsksetiska Nmnd; registration no N195/14) and were in accordance with The Guide for the Care and Use of Laboratory Animals (1996). During the investigation, every effort was made to minimize suffering and to reduce the number of animals used.

## Additional files

### Supplementary files
• Source code 1. Scripts.

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
