## [Decision Letter]

Thank you for submitting your article "Gaze shift neurons rely on the spatiotemporal interplay between multisensory excitation and recruited inhibition" for consideration by *eLife*. Your article has been reviewed by three peer reviewers, one of whom, Ronald L Calabrese, is a member of our Board of Reviewing Editors and the evaluation has been overseen by Timothy Behrens as the Senior Editor. The following individuals involved in review of your submission have agreed to reveal their identity: Herwig Baier (Reviewer #2); Mark Wallace (Reviewer #3).

The reviewers have discussed the reviews with one another and the Reviewing Editor has drafted this decision to help you prepare a revised submission.

Summary:

In this interesting study, the authors perform an electrophysiological and anatomical analysis of multimodal integration by excitatory output neurons and inhibitory GABAergic interneurons in the optic tectum (superior colliculus) of a primitive vertebrate, lampreys. This work uses several innovative preparations to record from output cells and interneurons both extracellularly and in whole cell mode while delivering biologically meaningful stimuli. They demonstrate that visual and electrosensory stimuli evoke direct excitation onto output neurons, which is quickly followed by recruited inhibition. These synaptic responses are enhanced when spatiotemporally aligned bimodal inputs are given, whereas temporally misaligned inputs are inhibited. They also show that multisensory inputs can regulate event-detection within tectum through this local inhibition. As in other vertebrates, there is evidence that in lampreys optic tectum output neurons commands gaze shifts by synaptic integration of different sensory modalities. Thus these findings have significant implications for how multimodal sensory inputs are integrated with behavioral responses in vertebrates.

Essential revisions:

The following major issues that should be addressed before publication in *eLife*.

1) Title is inappropriate: "Gaze shift neurons[]" Behavior is not studied here, and no direct link is shown between the cells characterized and gaze control. Please change title to reflect what is presented in the paper. The authors should also describe what shifts in gaze entail in lampreys: head movements, eye movements, etc. Consider adding lamprey to title.

2) Please do not over-generalize the results. a) The mechanisms described apply to a small sample of cells in the lamprey and should not be generalized to the entire vertebrate lineage. b) GABAergic tectal neurons are almost certainly not a homogenous population. While there is strong pharmacological evidence that some tectal interneurons are responsible for the observed feedforward inhibition, we are not convinced that this is generally true for all interneurons. In both cases the authors should tone down their claims.

3) "For statistical analysis, we used two-sample unpaired and paired t-tests in Matlab." We encourage the authors to use ANOVA followed by post-hoc testing that corrects for any multiple comparisons in each case where multiple treatments are made.

4) There was a general concern about the study and interpretation of temporal and spatial misalignment. a) In attempting to relate the temporal results to behavioral phenomena, what is the environmental range of latency differences for visual and electrosensory stimuli, and does this map on to the temporal effects demonstrated at the level of the individual neuron? b) The precipitous drop in EPSP amplitude (Figure 6) with very short latency differences between visual and electroreceptive stimulation is a bit surprising given the extensive literature in the mammalian SC of integration taking place over a fairly substantial window of time. Can these differences be readily reconciled? Furthermore, direct electrical stimulation of the retinal afferents to the OT bypasses all of the retinal processing that takes place, and that slows the conduction of information to the SC (relative to electroreception). Wouldn't one expect that peak gain would be seen when the visual inputs lag the electroreceptive inputs rather than at absolute simultaneity? c) In the experiments like those in Figure 1 it would seem appropriate to test temporally and also spatially misaligned stimuli. The authors never really study spatial misalignment, yet imply their results are relevant to such misalignment. Have such experiments been done and can they be incorporated?

5) The experiments of Figure 7 are intriguing but better quantification and consideration of the composite data is needed for the data of Figure 7. Just how selective is excitation to one quadrant of the retina within and across animals? Can this be quantified and statistically analyzed? How does it compare to inhibition?

[Editors' note: further revisions were requested prior to acceptance, as described below.]

Thank you for resubmitting your work entitled "Spatiotemporal interplay between multisensory excitation and recruited inhibition in the lamprey optic tectum" for further consideration at *eLife*. Your revised article has been favorably evaluated by Timothy Behrens (Senior Editor), a Reviewing Editor, and 2 reviewers.

The manuscript has been improved but there are some remaining issues that need to be addressed before acceptance, as outlined below:

Revisions have been extensive and responsive.

1) The new experiments with spatially misaligned stimuli (related to the new Figure 2) suggest that the source of inhibition is tectum-intrinsic. This is in disagreement with a previously published model (Mysore et al. 2013), based on experiments in other vertebrates. This work should be cited.

2) Panels 9E to 9G (new Figure 9) could use improvement. They are hard to understand. These panels are complex and difficult to follow (too many different colors). Also, why does Quadrant 1 (Q1) show higher PSP amplitudes? Please explain in legend or text more thoroughly.

3) Figure 9 legend: "from" is repeated. Please remove.

---

## [Author Response]

*Essential revisions:*

*The following major issues that should be addressed before publication in eLife.*

*1) Title is inappropriate: "Gaze shift neurons" Behavior is not studied here, and no direct link is shown between the cells characterized and gaze control. Please change title to reflect what is presented in the paper. The authors should also describe what shifts in gaze entail in lampreys: head movements, eye movements, etc. Consider adding lamprey to title.*

The title has been changed to: Spatiotemporal interplay between multisensory excitation and recruited inhibition in the lamprey optic tectum. The reason we originally included the term gaze shift neurons in the title is because they are tectal-brainstem projecting neurons and therefore implicated in the control of gaze movements. We have now removed gaze shift neurons from the title and have also incorporated our animal model.

Regarding gaze shifts in the lamprey, we agree with the referees that no behavior is studied in the present paper. However, a previous paper published from our lab describes in detail the kinematics and metrics of coordinated eye and head gaze shifts elicited from electrical microstimulation of the deep layer of the optic tectum in the lamprey (Saitoh et al., 2007). To better clarify what it is known in the lamprey, we have included the following sentence in the Introduction:

We have previously shown in the lamprey that site-specific stimulation across the deep layer of the optic tectum gives rise to eye-head gaze shifts of given amplitude and direction, thus, showing the existence of a motor map (Saitoh et al., 2007).

*2) Please do not over-generalize the results. a) The mechanisms described apply to a small sample of cells in the lamprey and should not be generalized to the entire vertebrate lineage. b) GABAergic tectal neurons are almost certainly not a homogenous population. While there is strong pharmacological evidence that some tectal interneurons are responsible for the observed feedforward inhibition, we are not convinced that this is generally true for all interneurons. In both cases the authors should tone down their claims.*

a) We agree that the results cannot be generalized without further evidence, although we think that the basic circuit is most likely very similar across vertebrates, given the present data and previous work in our lab, including evidence from mammals. To tone down our statements we have made the following modifications to the text:

1. We have changed the concluding sentence at the end of the Introduction (third paragraph) from:

[...] we anticipate that the mechanisms of integration of two senses on single output neurons, as demonstrated here, will also apply to other vertebrates. into:

[...] we anticipate that the mechanisms of integration of two senses on single output neurons, as demonstrated here, may also apply to other vertebrates.

2. We have changed the concluding sentence at the end of the Discussion (final paragraph) from:

We anticipate, without the loss of generality, that local inhibition is critical for spatiotemporal processing and that this mechanism may be found throughout vertebrate phylogeny into:

Local inhibition may be critical for spatiotemporal processing not only in lamprey but also in other vertebrates

b) We agree with the reviewers that not all superficial layer interneurons are necessarily causal to the generation of the feedforward inhibition onto tectal output neurons. For instance, it may very well be that a subset of interneurons may also be responsible for the intralaminar inhibition onto other interneurons as shown in Figure 8 and Figure 9. In our additional experiments, we further classified superficial interneurons into three provisional groups on the basis of their synaptic input (n=13) and, indeed, agree that they are not a homogenous population, suggesting that they may have a differential functional impact onto tectal multisensory integration. In fact, we find that some interneurons receive preferentially excitation and others excitation and inhibition we elaborate on this in point 5 and data shown in Figure 9.

To tone down this argument, we have deleted:

It is remarkable that both interneurons and the output neurons have an almost identical input, thus, suggesting that they are responsible for the feedforward inhibition elicited by spatiotopic sensory activation onto deep layer output neurons.

We have now replaced this with a concluding statement in the final paragraph of the Results section:

Despite the variable nature of visually-evoked synaptic inhibition onto tectal interneurons, the spatiotopic arrangement of their excitatory components is critical for the feed forward inhibition onto tectal output neurons.

However, we think the general strategy we propose for the inhibitory system is not affected by this heterogeneity, but instead we trust that this provides tectum with additional degrees of freedom for manipulating the sensorimotor processing, perhaps by other extrinsic sources (eg pallium/cortex). In order to highlight this in the manuscript, we have added the following sentence as a concluding sentence in the Discussion subsection Top-down regulation of the optic tectum by the forebrain:

The variable classes of tectal interneurons may also provide an additional flexibility to manipulate sensorimotor processing within the optic tectum.

*3) "For statistical analysis, we used two-sample unpaired and paired t-tests in Matlab." We encourage the authors to use ANOVA followed by post-hoc testing that corrects for any multiple comparisons in each case where multiple treatments are made.*

In most of our analyses, we compared one sensory modality (vision or electroreception) with their bimodal product (multisensory), and in these cases it is justified the use of a simple twoway statistical measure (t-test). The only case in which a multiple treatment was employed is with the voltage-clamp and current-clamp data shown in Figure 6 (former Figure 5). We appreciate the reviewers for pointing this out. We have now corrected the statistical technique by using a three-way ANOVA for illustrating that unisensory excitatory and inhibitory currents summate during bimodal integration. We have added the following text with values starting Results section, subsection Sensory excitation and local inhibition are fully integrated by output neurons:

"On a trial-to-trial basis, the statistical difference in the magnitude of evoked EPSC and IPSC currents by bimodal and visual or electroreceptive afferent stimulation was significant (visualbimodal: P = 0.008; electrosensory-bimodal: P = 0.006; one-way ANOVA), but not between those evoked during visual and electrosensory (P = 0.99).

[]

When bimodal inputs are used, excitatory currents summate and increase the resultant EPSP amplitude (Figure 5) but are quickly quenched by an also stronger amount of inhibition.''

4) There was a general concern about the study and interpretation of temporal and spatial misalignment. a) In attempting to relate the temporal results to behavioral phenomena, what is the environmental range of latency differences for visual and electrosensory stimuli, and does this map on to the temporal effects demonstrated at the level of the individual neuron? b) The precipitous drop in EPSP amplitude (Figure 6) with very short latency differences between visual and electroreceptive stimulation is a bit surprising given the extensive literature in the mammalian SC of integration taking place over a fairly substantial window of time. Can these differences be readily reconciled? Furthermore, direct electrical stimulation of the retinal afferents to the OT bypasses all of the retinal processing that takes place, and that slows the conduction of information to the SC (relative to electroreception). Wouldn't one expect that peak gain would be seen when the visual inputs lag the electroreceptive inputs rather than at absolute simultaneity? c) In the experiments like those in Figure 1 it would seem appropriate to test temporally and also spatially misaligned stimuli. The authors never really study spatial misalignment, yet imply their results are relevant to such misalignment. Have such experiments been done and can they be incorporated?

We break down our response to point 4 by dividing it into the spatial and time domain separately.

Spatial misalignment

To test the effectiveness of spatially disparate stimuli in the response suppression of the tectal output layer, we performed additional experiments by applying misaligned visual and electrosensory stimulation while locally recording in the optic tectum in the on-field region for one of the stimulus. These results we obtained are now summarized and shown in Figure 2. We have added the following text in the Results section (subsection Visual and electrosensory integration with local inhibtion):

Recently, we have shown how on-receptive field visual responses in tectal output layer neurons can be suppressed, via the local inhibitory system, by the presence of multiple visual stimuli located at disparate positions in the visual field (Kardamakis et al., 2015). [] An average response reduction of ~75% was observed during spatially misaligned cross-modal sensory stimulation (Figure 2; red for electrosensory and black for bimodal stimulation; n=5). By contrast, we were able to suppress by a negligible 5 amount of only ~3% (data not shown) when the opposite combination of sensory modalites were used, i.e. on-responses to visual and off-response to electrosensory stimuli. The underlying cause that gives rise to this asymmetry remains unclear.

These results show that responses to a given stimulus in on-field regions of tectum undergo a drastic suppression when stimulating off-field regions, thereby, triggering the local inhibitory system that aims to suppress areas across the tectal map of space. Surprisingly, we observed such a drastic reduction only when visual stimulation was applied to off-field areas while recording in an on-field region for a given electrosensory stimulus. However, we were not able to achieve response suppression when we reversed the order of sensory modalities, i.e. when we tried to get suppression of on-visual responses with off-field electrosensory stimulation (only very small reductions were observed, ~3%). We discuss our results and propose potential underlying reasons to explain this response suppression asymmetry in the next paragraph added to the Discussion (subsection The role of tectal inhibition):

This role of inhibition is observed in our extracellular recordings. Consistent with bimodal suppression in mammals (Meredith and Stein, 1996; Kadunce et al., 1997), responses in on-field regions of tectum to electrosensory stimulation undergo a drastic reduction when delivering visual stimuli to other receptive fields of the tectal map. In the present study, we could not elicit the same sort of reduction in tectal areas responsive for visual stimulation when applying offfield electrosensory stimuli. One possibility is that electrosensory maps in the lamprey tectum are not as refined as the visual ones, and therefore inhibition is less effective suppressing offregions.

In this sense, it has been shown that electrosensory receptive fields in elasmobranchs tectum are much larger than the visual ones (Bodznick, 1990). We cannot exclude either that the way we perform electrosensory stimulation is not as local as the visual one, making it less effective to evoke a similar reduction.

Temporal misalignment

Physiological responses to natural and localized bimodal stimuli with spatial offsets can not be obtained at the single neuron level using our half-brain preparation since natural/electrical topographic activation of sensory receptors would not be possible. However, in this current study, we exploited the fact that we can actually use whole-cell recordings to examine the net postsynaptic effect of temporal misalignments in response to stimulation of the two sensory afferent pathways. The advantage of using the latter approach is the fact that we obtained subthreshold membrane recordings during multisensory stimulation while quantifying the effects of tectal inhibition in integrating these responses, which was one of the major findings of this study.

In doing so, we uncovered that the bimodally-induced EPSPs elicited during temporally misaligned afferent activation reduced the amount of depolarization a fact predicted by the synaptic conductance estimations (Figure 6). The initial drop with 17% of the control observed in Figure 7 is significant but it should be noted that the overall depression lasts over 50 ms. The effects seen here are due to the direct stimulation of the afferent axons originating from the retina and the octavolateral relay center. This is clearly useful technically to elucidate the microcircuit but on the other hand it is not directly behaviorally relevant. To understand whether the modest temporal changes observed here are relevant in a behavioral context, we would need to record from a number of single isolated units simultaneously, while applying visual and electrosensory stimuli. This would be an important contribution, but represents in our mind a separate rather extensive future study, and is beyond the scope of this paper.

It is true that this window of integration appears longer in mammals (Meredith et al., 1987; Miller et al., 2015). One difference that emerges when comparing circuit dynamics between mouse SC and lamprey OT is that interlaminar inhibition mediating the superficial and deep layers is not monosynaptic, but rather polysynaptic. For instance, lamprey tectal output neurons are able to the discharge of one time-locked action potential in response to either sensory modality and this is a consequence of the fast suppression induced by the monosynaptic recruitment of local inhibition, thus, allowing for a narrow window of opportunity for membrane depolarization (see Figure 3, see also Kardamakis et al., 2015). In the mouse SC, microstimulation of the superficial layer or optic tract can evoke a burst of action potentials (see Saito Y. and Isa T., 2003). We attribute this discrepancy between the time-locked action potential found in lamprey OT and the burst of action potentials in mouse SC, in that local inhibition allows for a larger window of opportunity for multisensory inputs to summate for a longer duration, thus, enabling output neurons to discharge more action potentials since no monosynaptic IPSPs have been reportedly evoked from the superficial layer (see Saito Y. and Isa T., 2003). In addition to this, it still remains unknown to what extent neocortical involvement in the manipulation of the collicular GABAergic system may affect the window in mammals. Presumably, this recruited local inhibition is even further delayed, thus, enabling an even larger window of integration in cats and primates. For such a claim to be stated, however, more needs to be understood on the tectal microcircuit in cats/primates.

During natural conditions, the peak gain of bimodal stimulation would occur when electroreceptive inputs lag visual inputs due to the delay involved in the processing of visual information by the retina. However, electroreceptive inputs are also processed at the octavolateral nucleus delaying transmission to the OT at least by one synapse. In both our extracellular and whole-cell patch experiments, we never used the absolute simultaneity but rather always corrected for delays related in conduction of information. However, we agree that direct stimulation of the retina does indeed bypass retinal processing, thus, activating monosynaptically tectal-projecting optic ganglion cells. In a similar manner, when directly stimulating the octavolateral nucleus we are monosynaptically activating tectal-projecting electroreceptive fibers. We align the onsets of the evoked EPSPs by implementing a small delay (usually of the order of 1-5ms) and then perform temporally misaligned offsets to afferent stimulation. What the behavioral advantage of having a faster sensory system (electroreceptive in this case), and if this is true in natural conditions, remains a matter of speculation. Taken together, however, the tectal inhibitory system is equipped with the requisite connections to perform spatiotemporal computations.

5) The experiments of Figure 7 are intriguing but better quantification and consideration of the composite data is needed for the data of Figure 7. Just how selective is excitation to one quadrant of the retina within and across animals? Can this be quantified and statistically analyzed? How does it compare to inhibition?

To provide a better quantification and detailed explanation for this part, we performed additional experiments to increase the number (now 13 cells from 6 animals) of our recorded tectal interneurons by using our isolated eye-brain preparation. In doing so, we decided to divide the former Figure 7 into two figures (Figure 8 and Figure 9) with one focusing on the anatomy/morphology and the other on the electrophysiology. Now, Figure 9 illustrates the newly obtained data addressing the issues raised in this section:

a) By using whole-cell current clamp recordings in our eye-brain prep and QX314 in the recording pipette, we determined that all recorded interneurons responded retinotopically to mainly one quadrant. In most cases, they also responded weaker to microstimulation of the neighboring quadrant. As a measure of excitation and inhibition, the maximum PSP amplitude evoked in response to a brief stimulation train was used (Figure 8?). Quadrants (Q1, Q2, Q3 & Q4) are sorted on the basis of the maximal response obtained.

b) Prior to entering whole-cell configuration, we performed on-cell recordings in a subset of recorded interneurons (n=6) to determine their firing patterns by delivering a train of stimuli into their preferred quadrant. In all cases, we observed a time-locked action potential discharging to usually the first 2 stimuli with a subsequent depression and a rapid response to the recovery pulses. This is reminiscent of the discharge pattern of deep layer output neurons (as in Figure 3).

c) While interneurons receive retinotopic excitation, we saw that patterns of synaptic inhibition were, indeed, not identical. On this synaptic basis, we subdivided them into three provisional groups: i) local excitation-local inhibition (n=6, Figure 9), ii) local excitation global inhibition (n=4, Figure 9) and iii) local excitation-no inhibition (n=3, Figure 9).

d) The nature of this inhibitory component can be seen when switching into voltage-clamp (Figure 9), where recruitment of other interneurons would be required in order to generate this lateral inhibitory effect.

To account for all this new information, we modified Figure 8 to maintain only the anatomical, morphological and bimodal aspects of tectal interneurons and added Figure 9 to illustrate the variability of tectal interneurons on the basis of their topographically-driven synaptic input. We describe this at the end of the Results section in the manuscript in the last two paragraphs (subsection Bimodal sensory inputs also drive inhibitory interneurons):

To test if superficial layer interneurons are activated in a spatiotopic manner, like deep layer output cells, we used a preparation exposing the optic tectum to allow patch-clamp recordings from the superficial layer while keeping the retina and the optic nerve intact, so that local stimulation was possible (Figure 9, top; see Kardamakis et al., 2015 for experimental procedure).

[]

Despite the variable nature of visually-evoked synaptic inhibition onto tectal interneurons, the spatiotopic arrangement of their excitatory components is critical for the feedforward inhibition onto tectal output neurons.

[Editors' note: further revisions were requested prior to acceptance, as described below.]

1) The new experiments with spatially misaligned stimuli (related to the new Figure 2) suggest that the source of inhibition is tectum-intrinsic. This is in disagreement with a previously published model (Mysore et al. 2013), based on experiments in other vertebrates. This work should be cited.

As suggested by the reviewer, we have now included the reference Mysore et al. 2013.

To highlight this very important difference between endogenous and exogenous inhibition, we have added and modified a sentence Discussion section, subsection The role of tectal inhibition:

"While this local inhibition can act independently for generating stimulus selection, it has also been suggested that exogenous sources are required for generating global inhibition via GABAergic neurons in the isthmic nucleus (Mysore et al., 2013). Here, we show that inhibition hard-wired in the optic tectum via interneurons, that are activated from both retinal and electroreceptive afferents in the same way as the output neurons, is essential for multisensory stimulus selection."

2) Panels 9E to 9G (new Figure 9) could use improvement. They are hard to understand. These panels are complex and difficult to follow (too many different colors). Also, why does Quadrant 1 (Q1) show higher PSP amplitudes? Please explain in legend or text more thoroughly.

The reason we have chosen to use this wide variety of colors is that we chose to plot the 'actual' data obtained from individual interneurons and not the mean responses in order to highlight the differences between synaptic response profiles. Therefore, colors are necessary in order to distinguish which excitatory postsynaptic amplitude corresponded to which inhibitory postsynaptic amplitude in each recorded interneuron. To assist visualization of Figure 9, we have enlarged the panels by reducing the dead space on the x-axis, thickening the lines and increased some fonts.

To avoid dependencies on the particular retinal area (i.e., dorsal, anterior, ventral or posterior) that gives to the strongest excitatory component, quadrants are presented in descending order to facilitate the comparison of matching PSP across the population of interneurons that were recorded.

To clarify all of the above, we have added the following sentence of the figure legend: "However, to avoid dependencies on the particular stimulated retinal area, we sorted the quadrants that gives rise to the strongest excitatory component in descending order (Q1 being the largest EPSP). Lines with matching colors on each side of the y-axis represent the postsynaptic potential amplitudes obtained from individual recorded interneurons (positive values represent the total EPSP size and negative values represent the total IPSP size)."

3) Figure 9 legend: "from" is repeated. Please remove.

This has now been corrected.